# GOOD: Training-Free Guided Diffusion Sampling for Out-of-Distribution Detection

Xin Gao[1,2]*    Jiyao Liu[2]*    Guanghao Li[2]    Yueming Lyu[1]    Jianxiong Gao[2]    Weichen Yu[3]

Ningsheng Xu[2]    Liang Wang[4]    Caifeng Shan[1]    Ziwei Liu[5]    Chenyang Si[1,✉]

[1]Nanjing University    [2]Fudan University    [3]Carnegie Mellon University
[4]Chinese Academy of Sciences    [5]Nanyang Technological University

## Abstract

Recent advancements have explored text-to-image diffusion models for synthesizing out-of-distribution (OOD) samples, substantially enhancing the performance of OOD detection. However, existing approaches typically rely on perturbing text-conditioned embeddings, resulting in **semantic instability** and **insufficient shift diversity**, which limit generalization to realistic OOD. To address these challenges, we propose GOOD, a novel and flexible framework that directly guides diffusion sampling trajectories towards OOD regions using off-the-shelf in-distribution (ID) classifiers. GOOD incorporates dual-level guidance: (1) Image-level guidance based on the gradient of log partition to reduce input likelihood, drives samples toward low-density regions in pixel space. (2) Feature-level guidance, derived from k-NN distance in the classifier's latent space, promotes sampling in feature-sparse regions. Hence, this dual-guidance design enables more controllable and diverse OOD sample generation. Additionally, we introduce a unified OOD score that adaptively combines image and feature discrepancies, enhancing detection robustness. We perform thorough quantitative and qualitative analyses to evaluate the effectiveness of GOOD, demonstrating that training with samples generated by GOOD can notably enhance OOD detection performance.

## 1   Introduction

Out-of-distribution (OOD) detection is crucial for safely deploying machine learning systems, as neural networks often produce overly confident predictions when encountering distributional shifts [50, 69, 2, 1]. One promising approach to address this issue is Outlier Exposure (OE), which introduces auxiliary outlier datasets during training to help models learn more robust decision boundaries around in-distribution (ID) data [26, 33, 46, 48, 70]. Despite their effectiveness, OE methods depend heavily on manually selected outlier data, limiting their scalability and generalization.

Recent studies [36, 48, 7] have explored generative models as scalable alternatives to manually curated outlier datasets, with text-to-image diffusion models gaining attention for their ability to generate high-quality images. Typically, these methods follow a two-stage approach, first mapping ID samples into a latent space aligned with the text-conditional space of diffusion models, and then perturbing these embeddings to generate OOD samples [15, 13, 44]. However, this embedding process not only incurs high computational cost but also suffers from two fundamental limitations. First, as shown in Figure 1(a), small perturbations in the embedding space can result in disproportionate and unpredictable shifts in the image space through diffusion process. This misalignment hinders semantic control, often generating samples that either deviate significantly from the target class or remain indistinguishable from ID examples. Second, as illustrated in Figure 1(b), they primarily introduce

---

*Equal Contribution.

39th Conference on Neural Information Processing Systems (NeurIPS 2025).

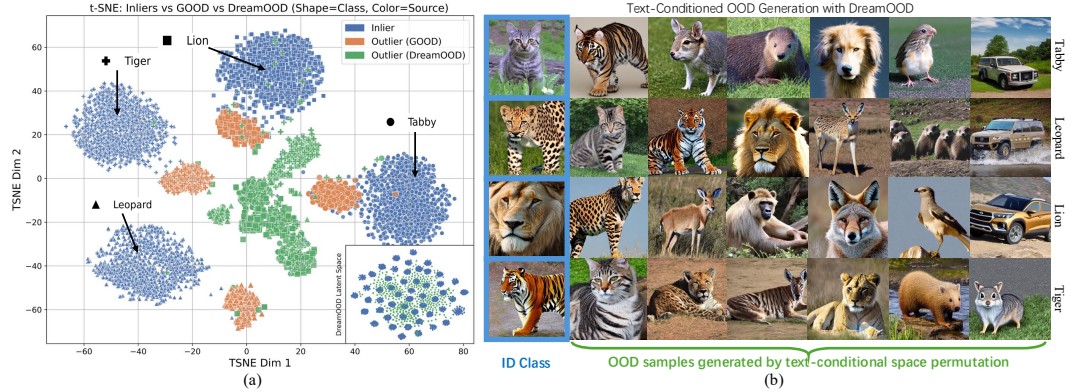

Figure 1: Limitations of text-conditioned OOD generation (e.g., Dream-OOD [15]). **Left: Misalignment** — Latent perturbations produce samples that either deviate from target semantics or resemble ID images. **Right: Limited Diversity** — Perturbing only semantic embeddings yields low coverage of broader distributional shifts, such as covariate or domain shift.

variation by shifting semantic embeddings, while neglecting other critical forms of distributional shift such as covariate and domain shifts [69, 6]. As a result, it becomes challenging to reliably generate diverse and informative OOD samples, which are essential for learning robust detectors.

To address these limitations, we aim to directly guide diffusion sampling in pixel space toward meaningful OOD regions instead of perturbing latent embeddings. Specifically, we leverage the ID classifier to provide distributional signals without additional training, inspired by the Training-Free Guidance (TFG) framework [71]. Unlike previous approaches that use differentiable functions primarily to ensure semantic or stylistic alignment [22, 56, 8, 4, 73], our goal is fundamentally different—to explicitly generate informative and diverse OOD samples to train a robust detector.

In this context, we propose GOOD, a framework for synthesizing OOD samples by explicitly guiding the denoising trajectory of diffusion models with ID classifier. GOOD introduces two guidance functions—image-level and feature-level—derived from differentiable OOD sores of any off-the-shelf classifier. The image-level guidance uses free energy to push samples toward low-likelihood regions in pixel space, while the feature-level guidance leverages k-nearest neighbor distances in the feature space to promote sampling in feature-sparse areas. Together, these signals enable the generation of diverse and informative OOD samples that are clearly separated from the ID distribution. We also analyze the diffusion process and show that proper initialization and step-size control are key to effective boundary sampling. These samples are then used in an OE setting to enhance classifier robustness. Additionally, we introduce a unified OOD score combining pixel-level likelihood and feature-space distance, which improves detection across various OOD scenarios.

We conduct extensive experiments on multiple benchmarks, including Imagenet, CIFAR-100, CIFAR-10, to evaluate the effectiveness of GOOD. Results show that incorporating GOOD-generated samples into OE training significantly improves detection performance, outperforming existing post-hoc and synthesized outlier-based methods. Our contributions are summarized as follows:

- We propose GOOD, a flexible framework that guides diffusion sampling toward OOD regions using off-the-shelf classifiers. The synthesized outliers effectively encourage classifiers to establish conservative and robust decision boundary for improved OOD detection.

- Additionally, we propose an unified OOD Score, which integrates pixel-level and feature-level discrepancies, providing a more comprehensive measure for OOD detection.

- Extensive experiments across benchmark datasets showing that GOOD outperforms previous OOD generation and OE methods in terms of detection accuracy.

## 2 Preliminaries and Related Works

We consider a training set $D = \{(x_i, y_i)\}_{i=1}^n$, drawn *i.i.d.* from a joint distribution $P_{\mathcal{X}\mathcal{Y}}$, where $x_i \in \mathcal{X}$ and $y_i \in \mathcal{Y} = \{1, \ldots, C\}$. Let $\mathbb{P}_{in}$ be the marginal input distribution, representing the

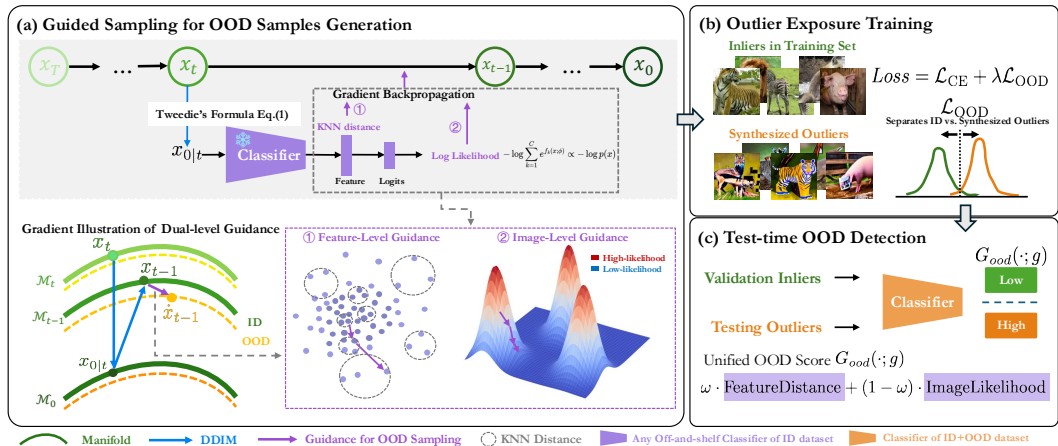

Figure 2: Overview of the GOOD framework. (a) Guided by feature- and image-level signals, GOOD samples synthetic outliers near the decision boundary. (b) These outliers are used in Outlier Exposure training to separate ID and OOD samples via a regularization loss. (c) At test time, OOD is detected using a unified score that combines feature distance and image likelihood.

*in-distribution* (ID). We define a multi-class classifier $f_\phi : \mathcal{X} \mapsto \mathbb{R}^C$, parameterized by $\phi$, which predicts the label of each input sample.

**Out-of-distribution Detection.** In real-world applications, a reliable classifier must not only make accurate predictions on *in-distribution* (ID) data, but also identify *out-of-distribution* (OOD) inputs that differ significantly from the training distribution. Such shifts may stem from changes in domain, covariates, or semantics [69, 6], rendering OOD samples incompatible with the label space $\mathcal{Y}$. To separate ID from OOD samples, an ***OOD score function*** $G_{ood}(x) : \mathcal{X} \mapsto \mathbb{R}$ is used, where ID samples receive lower scores and OOD samples higher ones. This score is typically based on model confidence [25], likelihood estimates [46, 59], or distance-based metrics [61, 38].

**Generative Diffusion Model.** Diffusion models generate data by reversing a forward noising process. Given a sample $x_0 \sim p_0(x)$, the forward (diffusion) process gradually corrupts $x_0$ into a noisy sample $x_t$ over discrete time steps $t \in [T]$: $x_t = \sqrt{\bar{\alpha}_t}x_0 + \sqrt{1 - \bar{\alpha}_t}\epsilon$, $\quad \epsilon \sim \mathcal{N}(0, I)$, where $\{\bar{\alpha}_t\}_{t=1}^T$ defines the noise schedule. The denoising model $\epsilon_\theta(x_t, t)$, parameterized by $\theta$, is trained to predict the added noise $\epsilon$. This yields the standard noise prediction objective for training proposed in [27].

For sampling, we begin with $x_T \sim N(0, I)$ and iteratively sample $x_{t-1} \sim p_{t-1|t}(x_{t-1}|x_t)$. Since this conditional probability is not directly computable, DDIM [55] approximates $x_{t-1}$ as:

$$x_{t-1} = \sqrt{\bar{\alpha}_{t-1}}x_{0|t} + \sqrt{1 - \bar{\alpha}_{t-1} - \sigma_t^2}\frac{x_t - \sqrt{\bar{\alpha}_t}x_{0|t}}{\sqrt{1 - \bar{\alpha}_t}} + \sigma_t\epsilon, \quad x_{0|t} = m(x_t) \triangleq \frac{x_t - \sqrt{1 - \bar{\alpha}_t}\epsilon_\theta(x_t, t)}{\sqrt{\bar{\alpha}_t}}, \quad (1)$$

where $\sigma_t$ controls the stochasticity of the sampling process, and $x_{0|t}$ is the model's estimate of the original signal at step $t$, given by Tweedie's formula [17].

For diffusion models such as Stable Diffusion (SD) [53], which generate data from a conditional distribution $p_0(x|c)$, the condition $c$ typically lies in a *text-conditional space*, enabling control over generation via descriptions. These models are trained to predict $\epsilon_\theta(x_t, c, t)$, where $c$ may be a descriptive text or left empty. During sampling, guidance is applied by modifying the predicted noise to steer the output more strongly toward the conditioning signal [29, 52]:

$$\hat{\epsilon}_\theta(x_t, c, t) = (1 + \beta)\epsilon_\theta(x_t, c, t) - \beta\epsilon_\theta(x_t, \varnothing, t), \quad (2)$$

where $\beta$ controls the strength of conditioning.

We have briefly introduced some relevant works in **OOD detection** and **guided diffusion sampling** as context for our method. For a comprehensive review of related literature, please refer to Appendix A.1.

## 3 GOOD: Guided OOD Sampling with Diffusion Models

As shown in Figure 2, our framework GOOD enhances OOD detection by generating informative synthetic outliers for Outlier Exposure training. Section 3.1 introduces two guidance—feature- and

---

**Algorithm 1** Training-Free Guidance for OOD Sampling (GOOD)

---

1: **Input:** Diffusion model $\epsilon_\theta(\cdot, \cdot, \cdot)$, condition $c$, condition strength $\beta$, ID Classifier $f_\phi$, k-NN distance $k$, guidance strength $\rho, \mu, \bar{\gamma}$, number of steps $T$
2: $x_T \sim \mathcal{N}(0, I)$
3: **for** $t = T, \ldots, 1$ **do**
4:     Conditional Predicted Noise $\hat{\epsilon}_\theta(x_t, c, t) = (1 + \beta)\epsilon_\theta(x_t, c, t) - \beta\hat{\epsilon}_\theta(x_t, \emptyset, t)$
5:     Define function $\tilde{f}(x) = \mathbb{E}_{\delta \sim \mathcal{N}(0, I)} f(x + \bar{\gamma}\sqrt{1 - \bar{\alpha}_t}\delta)$                     $\triangleright$ TFG (a)
6:     Guided function for OOD Sampling $G_{ood}(\cdot; \tilde{f}) = E(\cdot; \tilde{f})$ or $D_k(\cdot; \tilde{f})$
7:     $x_{0|t} = \left(x_t - \sqrt{1 - \bar{\alpha}_t}\hat{\epsilon}_\theta(x_t, c, t)\right)/\sqrt{\bar{\alpha}_t}$           $\triangleright$ Obtain the predicted data
8:     $\Delta_t = \rho_t \nabla_{x_t} G_{ood}(x_{0|t}; \tilde{f})$                            $\triangleright$ TFG (b)
9:     $\Delta_0 = \mu_t \nabla_{x_{0|t}} G_{ood}(x_{0|t}; \tilde{f})$                          $\triangleright$ TFG (c)
10:     $x_{t-1} = \text{Sample}(x_t, x_{0|t}, t) + \Delta_t/\sqrt{\alpha_t} + \sqrt{\bar{\alpha}_{t-1}}\Delta_0$     $\triangleright$ Sample as DDIM in Eq. (1)
11:     $x_t \sim \mathcal{N}(\sqrt{\alpha_t}x_{t-1}, \sqrt{1 - \alpha_t}I)$
12: **end for**
13: **Output:** OOD sample $x_0$ with respect to $f_\phi$

---

image-level—using an off-the-shelf classifier to guide sampling toward OOD regions. Section 3.2 refines the sampling trajectory by adjusting step size and initialization to produce outliers with varying anomaly levels near the decision boundary. These are then used to boost model robustness. Finally, Section 3.3 proposes a unified OOD scoring function for effective detection.

## 3.1 Two Training-Free Guidance for OOD Sampling

To explore how diffusion models can generate OOD samples, we visualize their sampling trajectories with gradient cues in Figure 2. In particular, we examine the transition from the noisy manifold $\mathcal{M}_t$ to $\mathcal{M}_{t-1}$ (shown by the blue arrow) and observe that it can be guided toward OOD regions using gradients from an ID classifier. Motivated by this observation, we introduce two complementary guidance mechanisms—**image-level guidance** (GOOD$_{img}$) and **feature-level guidance** (GOOD$_{feat}$)—which leverage both likelihood and distance. These guidances can be derived from any off-the-shelf classifier, making our approach both model-agnostic and training-free.

**Image-Level Guidance: GOOD$_{img}$.** Following the energy-based reinterpretation of discriminative classifiers [19], we *define* an energy for a classifier with logits $f_y(x; \phi)$ as

$$E_\phi(x, y) := -f_y(x; \phi), \qquad E_\phi(x) := -\log\sum_{k=1}^{C} e^{f_k(x; \phi)}. \tag{3}$$

These definitions induce the model-dependent joint and marginal densities

$$p_\phi(x, y) = \frac{e^{-E_\phi(x, y)}}{Z(\phi)}, \qquad p_\phi(x) = \sum_y p_\phi(x, y) = \frac{\sum_k e^{f_k(x; \phi)}}{Z(\phi)} = \frac{e^{-E_\phi(x)}}{Z(\phi)}, \tag{4}$$

where $Z(\phi)$ is a parameter-dependent partition function constant in $x$. Hence $E_\phi(x)$ is a *model-defined negative log-density* (up to $Z(\phi)$), sometimes called the *free energy* [35, 46]. Intuitively, samples with larger $p_\phi(x)$ (i.e., lower energy) correspond to regions the classifier assigns high confidence to; lower $p_\phi(x)$ indicates atypical or OOD regions. Following [46], in-distribution samples tend to concentrate in high-$p_\phi(x)$ (low-energy) regions, as supported by their gradient-based analysis. We thus define the image-level guidance gradient as

$$\nabla_x E_\phi(x) = -\frac{1}{\sum_k e^{f_k(x; \phi)}} \sum_k e^{f_k(x; \phi)} \nabla_x f_k(x; \phi). \tag{5}$$

During reverse diffusion, moving along $+\nabla_x E_\phi(x)$ pushes samples from high-density (ID) toward low-density (OOD) regions in the input space of $f_\phi$.

**Feature-Level Guidance: GOOD$_{feat}$** The feature space learned by deep networks is typically lower-dimensional, more compact, and semantically richer than the raw image space. To exploit this property, we define feature-level OOD guidance using the k-nearest neighbor (k-NN) distance in the

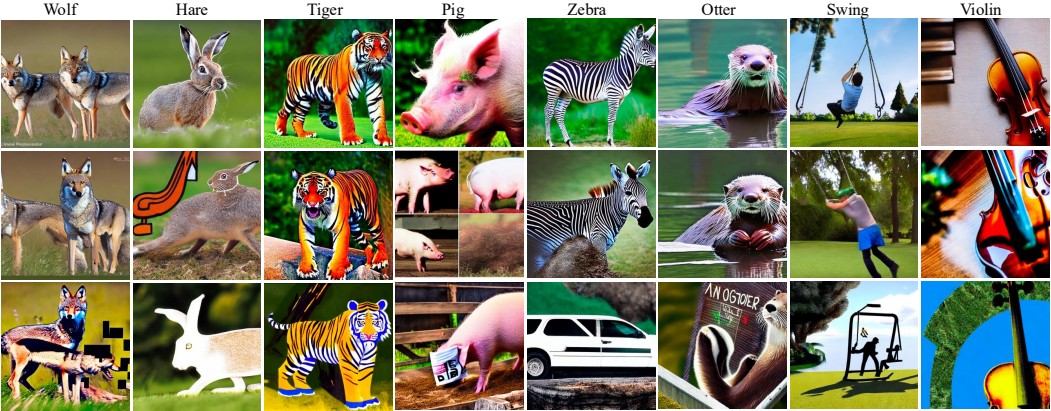

| Wolf | Hare | Tiger | Pig | Zebra | Otter | Swing | Violin |

Figure 3: Visualization of generated OOD samples with increasing step sizes, arranged **from top to bottom** ($\rho = \mu = 0.1 \rightarrow 2 \rightarrow 5$). The samples exhibit varying degrees of anomaly.

feature space. Let $f^{1:L}(x; \phi) = z$ denote the $L$-layer feature representation of input $x$. We construct a normalized embedding bank $\mathbb{Z} = \{z_1, z_2, \dots, z_n\}$, where each $z_i = f^{1:L}(x_i; \phi) / \|f^{1:L}(x_i; \phi)\|_2$.

For a given embedding $z$ from input $x$, its k-NN distance to the embeddings in $\mathbb{Z}$ is:

$$D_k(x, f) \triangleq \|z - z_{(k)}\|_2 = \|\tfrac{f^{1:L}(x;\phi)}{\|f^{1:L}(x;\phi)\|_2} - z_{(k)}\|_2, \tag{6}$$

where $z_{(k)}$ is the $k$-th nearest neighbor of $z$ in $\mathbb{Z}$. Intuitively, embeddings located in sparse regions (i.e., large k-NN distance) are more likely to be near the distribution boundary or truly OOD, while those in dense regions are likely ID. We define feature-level guidance GOOD$_{feat}$ as the gradient of this k-NN distance with respect to the input:

$$\nabla_x D_k(x; f) = \tfrac{2}{\|f^{1:L}(x_i;\phi)\|_2} \cdot \left[ \tfrac{f^{1:L}(x;\phi)}{\|f^{1:L}(x_i;\phi)\|_2} - z_{(k)} \right] \cdot \nabla_x f^{1:L}(x; \phi). \tag{7}$$

Along this gradient direction, we generate samples with increasingly rarer features, as depicted by the transition from smaller to larger circles in part (b) of Figure 2.

While both GOOD$_{img}$ and GOOD$_{feat}$ are defined on clean inputs $x_0$, the diffusion sampling process operates on the noised input $x_t$. To bridge this gap effectively, we adopt the **Training-Free Guidance** (TFG) framework [71], which unifies existing training-free sampling strategies and allows their combination toward any differentiable target predictor. The overall sampling algorithm is summarized in Algorithm 1. GOOD can be seamlessly integrated into the TFG framework, with a detailed analysis provided in Appendix. This approach enables the application of GOOD$_{img}$ and GOOD$_{feat}$ throughout the reverse diffusion process without the need for retraining the model.

## 3.2 Balanced OOD Sampling with Varying Degrees of Anomaly

Building on the differentiable OOD score from Section 3.1, our method captures diverse distributional shifts without manual definitions, aligning with the model's implicit understanding of OOD. However, while gradients guide the direction, the target remains uncontrolled, which may limit the algorithm's application. To improve this, we refine the process by adjusting the initial point and step size, enabling the generation of OOD samples with varying levels of anomaly.

**Initialization via Conditional Generation.** Unconditional generation can cause samples to deviate significantly from the original distribution due to the broad coverage of the generative model, failing to properly regularize the classifier. To mitigate this, we leverage diffusion models like SD, which can incorporate class descriptions (e.g., "A high-quality image of the <class label>") as conditional inputs. This essentially selects an initial point near the ID class, with the gradient guiding the process further from the distribution center, generating samples with increasing levels of anomaly.

**Step Size of Different Combinations.** The hyperparameters $\rho$ and $\mu$ in Algorithm 1 directly control the step size of the sampling trajectory and thereby influence the characteristics of OOD samples. In the TFG framework, grid search is used to identify the optimal combination based on the visual quality of the generated images. To better understand how step size affects the usefulness of samples for OOD detection, we perform an empirical analysis using GOOD$_{img}$ defined in Eq. (3).

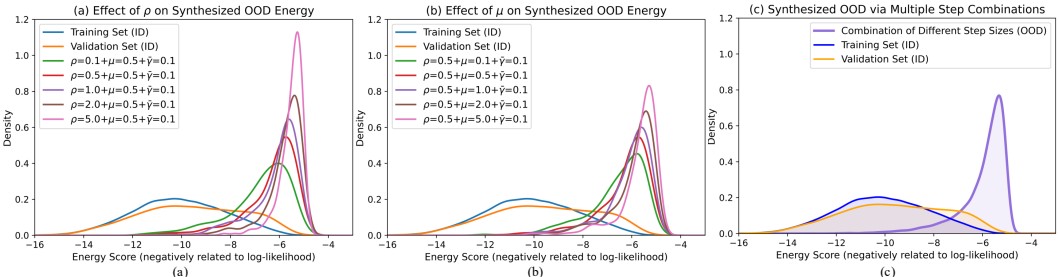

Figure 4: Energy score analysis (inversely related to image likelihood) on ImageNet-100 ID sets and synthesized OOD samples. (a) OOD energy distributions across different $\rho$ values. (b) Distributions under varying $\mu$. (c) Balanced sampling over all $\rho$–$\mu$ combinations ($5 \times 5 = 25$).

As shown in Figure 3, increasing the step size (from top to bottom) leads to samples that diverge more noticeably in style, domain, and semantics. Figures 4(a) and (b) further provide a statistical view: the energy scores of generated samples increasingly diverge from those of the ID validation set as step size grows. The distributions shift rightward and become more peaked, with $\rho$ having a stronger impact than $\mu$. This is likely because $\rho$ incorporates second-order variance information, enabling more accurate gradient directions and producing more concentrated sampling distributions.

Samples that are too close to the ID distribution may be indistinguishable from ID data, weakening their regularization effect. Conversely, samples that deviate too far are too easy to separate, resulting in overly loose decision boundaries. To address this, we propose combining samples generated with multiple parameter settings rather than relying on a single "optimal" configuration. This produces a more realistic near-OOD distribution and promotes tighter, more robust decision boundaries, as shown in Figure 4(c) and detailed in Appendix.

**Outlier Exposure with Synthesized OOD data.** The generated OOD samples are subsequently used for OE training to improve the classifier's ability to distinguish between ID and OOD data. The OOD loss is defined as [16]:

$$\mathcal{L}_{\text{OOD}} = \mathbb{E}_{x_{\text{OOD}}} \left[ -\log \frac{1}{1 + \exp \psi(E(f_\phi(x_{\text{OOD}})))} \right] + \mathbb{E}_{x \sim P_{\text{in}}} \left[ -\log \frac{\exp \psi(E(f_\phi(x)))}{1 + \exp \psi(E(f_\phi(x)))} \right], \tag{8}$$

where $\psi(\cdot)$ is a three-layer MLP trained to predict OOD likelihood, $E(\cdot)$ denotes the *free energy* in Eq. (3). The final loss function is $\mathcal{L} = \mathcal{L}_{\text{CE}} + \lambda \cdot \mathcal{L}_{\text{OOD}}$, where $\lambda$ is a hyperparameter controlling the regularization strength. $\mathcal{L}_{\text{CE}}$ denotes the cross-entropy loss on the ID training data. By incorporating both ID and synthesized OOD samples into training, the classifier learns to better differentiate their distribution, thereby improving its generalization to unseen data in real-world scenarios.

### 3.3   Combining Image Likelihood and Feature Distance for Unified OOD Scoring

To improve OOD detection during testing, we propose a **unified OOD score** that combines two complementary signals: image-level likelihood and feature-level distance.

The free energy $E(x; \phi)$, defined in Eq. (3), represents the negative log-likelihood of an input image, reflecting how well it fits the learned image distribution. While it captures pixel-level shifts effectively, it may struggle to distinguish true OOD samples from ID inputs in low-density regions due to the high dimensionality and sparsity of image space. In contrast, the k-NN distance $D(x; \phi)$, defined in Eq. (6), measures compactness in a lower-dimensional feature space, making it better suited for detecting class and feature anomalies.

Instead of favoring one metric, we treat them as complementary, combining them to improve OOD detection across different scenarios. To adaptively weight the two signals, we estimate the distributional shift in feature space by comparing the k-NN distance distributions of the test set and a held-out ID validation set. Let $q_{\text{test}}$ and $q_{\text{ID}}$ denote their empirical distributions, and compute the Kullback–Leibler (KL) divergence: $\text{KL}(q_{\text{test}} \| q_{\text{ID}})$. A larger divergence indicates significant deviation in feature space, suggesting higher reliability of the feature-level score. Conversely, a smaller divergence favors the image-level score. Thus, we define a soft interpolation coefficient: $w = 1 - \exp(-a \cdot \text{KL}(q_{\text{test}} \| q_{\text{ID}})), a > 0$, which increases with the degree of shift and smoothly maps to the range $[0, 1]$. The final unified OOD score for a test sample $x$ is computed as:

$$G_{ood}(x; f) = w \cdot D(x; f) + (1 - w) \cdot \hat{E}(x; f), \tag{9}$$

Table 1: Comparative evaluation of OOD detection performance on ImageNet-100 as the in-distribution dataset (following Dream-OOD [15]). We report standard deviations estimated across 3 runs. Bold numbers are superior results and underline indicates second best.

| Methods | INATURALIST | | PLACES | | SUN | | TEXTURES | | Average | | ID Acc |
|---|---|---|---|---|---|---|---|---|---|---|---|
| | FPR95↓ | AUROC↑ | FPR95↓ | AUROC↑ | FPR95↓ | AUROC↑ | FPR95↓ | AUROC↑ | FPR95↓ | AUROC↑ | |
| MSP [25] | 31.80 | 94.98 | 47.10 | 90.84 | 47.60 | 90.86 | 65.80 | 83.34 | 48.08 | 90.01 | 87.64 |
| ODIN [43] | 24.40 | 95.92 | 50.30 | 90.20 | 44.90 | 91.55 | 61.00 | 81.37 | 45.15 | 89.76 | 87.64 |
| Mahalanobis [38] | 91.60 | 75.16 | 96.70 | 60.87 | 97.40 | 62.23 | 36.50 | 91.43 | 80.55 | 72.42 | 87.64 |
| Energy [46] | 32.50 | 94.82 | 50.80 | 90.76 | 47.60 | 91.71 | 63.80 | 80.54 | 48.68 | 89.46 | 87.64 |
| G-ODIN [30] | 39.90 | 93.94 | 59.70 | 89.20 | 58.70 | 90.65 | 39.90 | 92.71 | 49.55 | 91.62 | 87.38 |
| kNN [61] | 28.67 | 95.57 | 65.83 | 88.72 | 58.08 | 90.17 | 12.92 | 90.37 | 41.38 | 91.20 | 87.64 |
| ViM [64] | 75.50 | 87.18 | 88.30 | 81.25 | 88.70 | 81.37 | 15.60 | 96.63 | 67.03 | 86.61 | 87.64 |
| ReAct [59] | 22.40 | 96.05 | 45.10 | 92.28 | 37.90 | 93.04 | 59.30 | 85.19 | 41.17 | 91.64 | 87.64 |
| DICE [60] | 37.30 | 92.51 | 53.80 | 87.75 | 45.60 | 89.21 | 50.00 | 83.27 | 46.67 | 88.19 | 87.64 |
| *Synthesis methods* | | | | | | | | | | | |
| GAN [37] | 83.10 | 71.35 | 83.20 | 69.85 | 84.40 | 67.56 | 91.00 | 59.16 | 85.42 | 66.98 | 79.52 |
| VOS [16] | 43.00 | 93.77 | 47.60 | 91.77 | 39.40 | 93.17 | 66.10 | 81.42 | 49.02 | 90.03 | 87.50 |
| NPOS [62] | 53.84 | 86.52 | 59.66 | 83.50 | 53.54 | 87.99 | 8.98 | 98.13 | 44.00 | 89.04 | 85.37 |
| Dream-OOD [15] | 24.10 | 96.10 | 39.87 | 93.11 | 36.88 | 93.31 | 53.99 | 85.56 | 38.76 | 92.02 | 87.54 |
| NCIS [13] | 20.70 | 96.56 | 34.60 | 94.07 | 35.43 | 94.13 | 44.83 | 88.50 | 33.89 | 93.32 | 87.24 |
| BOOD [44] | 18.33 | 96.74 | 33.33 | 94.08 | 37.92 | 93.52 | 51.88 | 85.41 | 35.37 | 92.44 | 87.92 |
| **GOOD (Ours)** | **9.22**±0.3 | **97.61**±0.2 | **24.79**±0.4 | **94.82**±0.1 | **17.20**±1.0 | **96.10**±0.2 | **19.51**±1.1 | **96.60**±0.1 | **17.68**±0.7 | **96.30**±0.2 | 87.16 |

Table 2: Comparative evaluation of OOD detection performance on CIFAR-100 as the ID dataset.

| Methods | SVHN | | PLACES265 | | LSUN-R | | ISUN | | TXETURES | | Average | |
|---|---|---|---|---|---|---|---|---|---|---|---|---|
| | FPR95↓ | AUROC↑ | FPR95↓ | AUROC↑ | FPR95↓ | AUROC↑ | FPR95↓ | AUROC↑ | FPR95↓ | AUROC↑ | FPR95↓ | AUROC↑ |
| w/o OOD | 48.29 | 84.38 | 65.24 | 84.38 | 42.82 | 86.62 | 31.24 | 89.19 | 53.97 | 83.07 | 48.31 | 84.21 |
| Dream-OOD [15] | 58.75 | 87.01 | 70.85 | 79.94 | 24.25 | 95.23 | 1.10 | 99.73 | 46.60 | 88.82 | 40.31 | 90.15 |
| FodFoM [7] | 33.19 | 94.02 | **42.30** | 90.68 | 28.24 | 95.09 | 33.06 | 94.45 | 35.44 | 93.38 | 34.45 | 93.52 |
| GOOD (SD 512*512) | 19.37 | 95.34 | 65.85 | 82.04 | 18.82 | 94.15 | 0.81 | 99.59 | **23.47** | 95.98 | 25.66 | 93.34 |
| GOOD (SD 128*128) | **6.65** | **98.62** | 54.92 | 82.09 | **6.09** | **98.73** | **0.07** | **99.96** | 29.79 | 94.01 | **19.50** | **94.68** |

where $\hat{E}(x; \phi)$ is min-max normalized to $[0, 1]$ for consistency during interpolation.

# 4 Experiment

We provide a comprehensive evaluation of our proposed framework GOOD. GOOD generates meaningful pixel-level outliers that exhibit distributional shifts from ID data, resulting in significant improvements in OOD detection on ImageNet [10] and CIFAR-100 [34] (Section 4.2). Ablation studies (Section 4.3) are conducted to assess the impact of various components and hyperparameters.

## 4.1 Experimental Setup

**Dataset and Training Details.** We use ImageNet-100 [10] and CIFAR-100 as the ID training datasets, following [15, 13, 44, 7]. For ImageNet-100, the OOD test data is sourced from [32], which includes iNaturalist [63], SUN [66], Places [74], and Textures [9]. For CIFAR-100, we evaluate on five OOD test sets: SVHN [49], Places365 [74], LSUN-R [72], ISUN [67], and Textures [9].

Following [15], we use ResNet-34 [21] as the ID classifier for fair comparison, though our method is architecture-agnostic and works with any pretrained model on the ID dataset. To improve the classifier's ability to differentiate between OOD and ID data, we generate OOD samples using Stable Diffusion v1.5 [52] guided by our proposed method. Specifically, we generate 12,500 images using image-level guidance ($GOOD_{img}$) and 10,500 images using feature-level guidance ($GOOD_{feat}$). The detection model is fine-tuned from the pretrained classifier using these synthetic OOD samples and the loss function in Eq. (8). The loss weighting parameter $\lambda$ is 2.5. Optimization is performed using stochastic gradient descent with momentum (0.9) and a weight decay of $5 \times 10^{-4}$. The model is trained for 50 epochs on ImageNet-100 and 200 epochs on CIFAR-100, with a cosine learning rate schedule that starts with an initial learning rate of $10^{-4}$ and a batch size of 160.

**Evaluation Metrics.** We evaluate OOD detection performance using three metrics: (1) FPR95 — false positive rate when the true positive rate for ID samples is 95%; (2) AUROC — area under the receiver operating characteristic (ROC) curve; and (3) ID Acc — ID classification accuracy.

## 4.2 Evaluation of OOD Detection Performance and Comparative Analysis

**GOOD significantly improve OOD detection.** Table 1 presents a comprehensive comparison of our method, GOOD, against a diverse set of baselines, including score-based methods (e.g., Maximum

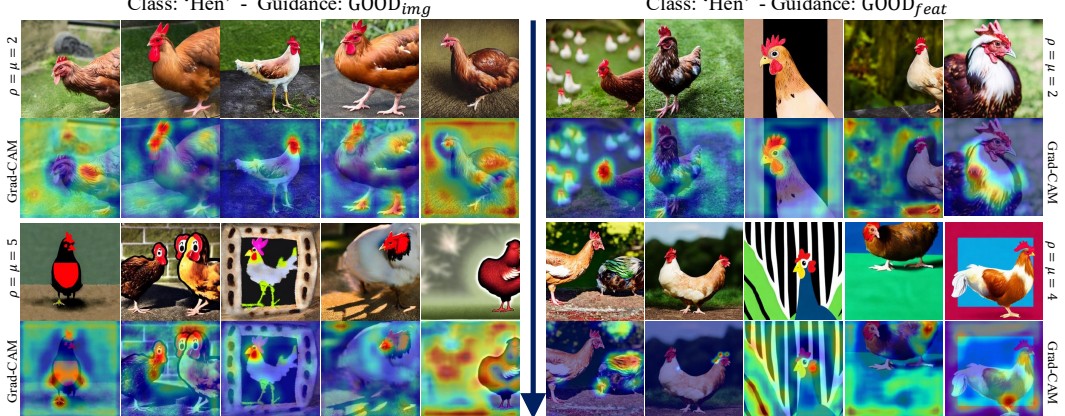

Figure 5: Generated OOD samples using two guidance types, with Grad-CAM heatmaps highlighting regions of high gradient response. The class 'hen' is from ImageNet-100.

Softmax Probability [25], ODIN [43], Mahalanobis [38], Energy Score [46], Generalized ODIN [30], KNN Distance [61], ViM [64], ReAct [59], DICE [60]) and synthesis-based approaches (e.g., VOS [16], NPOS [62], GAN-based generation [37]). We also include recent diffusion-based methods closely related to our work, such as DreamOOD [15], NCIS [13], BOOD [44], and FodFom [7].

On the ImageNet-100 dataset, GOOD significantly outperforms existing methods, achieving the lowest FPR95 (17.68) and highest AUROC (96.30), clearly surpassing the second-best method, NCIS. This demonstrates its robustness in challenging OOD scenarios such as iNaturalist, Places, and SUN. On the Textures dataset, GOOD slightly trails NPOS, which operates in the latent space. Notably, all pixel-level synthesis methods struggle on Textures. As detailed in the Appendix, this dataset is easily separable from ID data at the feature level, but Stable Diffusion struggles to generate OOD images with similar features due to its training distribution. GOOD mitigates this by introducing feature-level guidance during generation and combining image and feature cues at test time. This limitation can be mitigated with more powerful generative models that capture broader distributions, and our method offers a flexible framework adaptable to future advancements.

**GOOD adapts with smaller diffusion models and low-resolution datasets.** Table 2 compares GOOD using 512×512 and 128×128 diffusion models. Prior methods often use high-res models (e.g., Stable Diffusion at 512×512) on low-res datasets like CIFAR-100, causing mismatches due to up/downsampling, which misaligns pixel and feature spaces and degrades OOD image quality.

GOOD avoids this by directly guiding the denoising process via gradients. We use a high-res generator and apply differentiable bilinear downsampling to match the classifier's 32×32 input. This fully differentiable path allows gradients to emphasize key regions, producing meaningful OOD images. With a 128×128 model, GOOD reduces average FPR95 from 34.5% (FodFoM) to 19.50%, and improves AUROC from 93.52 to 94.68. Gains are especially notable on ISUN (0.07% FPR95, 99.96 AUROC) and SVHN (6.65% FPR95, 98.62 AUROC). The 512×512 model only outperforms on the highly textured TEXTURES set (23.47% FPR95), suggesting high detail helps in specific cases. Full results are in Appendix. Overall, the smaller 128×128 model consistently outperforms its larger counterpart—likely due to a better inductive bias for low-resolution structure and the elimination of information loss during resizing—while requiring an order-of-magnitude less compute.

## 4.3 Ablation Study

In this section, we present additional ablation studies to further analyze the effectiveness of GOOD in generating informative OOD samples and improving detection performance.

**OOD emerges from classifier-critical regions.** Figure 5 shows OOD samples generated by our two guidance strategies: $GOOD_{img}$ (left) and $GOOD_{feat}$ (right). Each image is paired with a Grad-CAM heatmap, highlighting regions most sensitive to the OOD score. As previously shown in Figure 4, the guided samples are well-separated from ID data in distribution. The visualizations now reveal how this separation emerges: although the generated images remain close to the ID class (e.g., hens), they exhibit subtle yet semantically meaningful deviations. These shifts often occur in key semantic

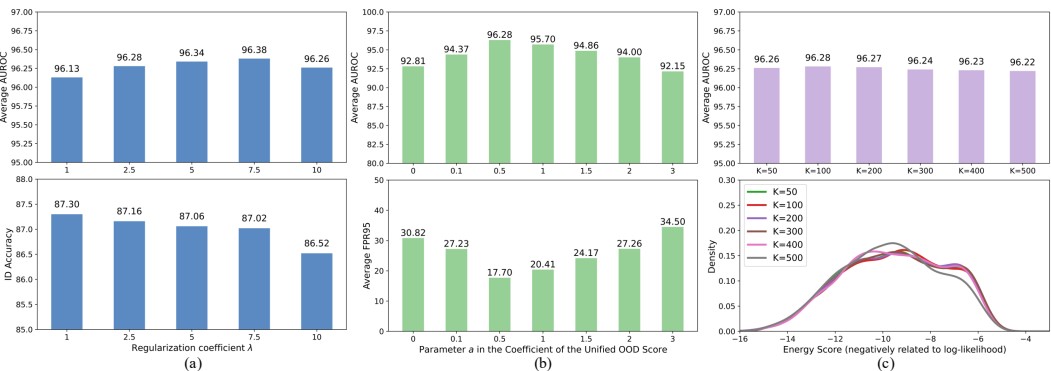

Figure 6: Ablation studies on key hyperparameters: (a) regularization weight $\lambda$ in the OOD loss $\mathcal{L}_{\text{ood}}$; (b) interpolation coefficient $a$ in the unified OOD score (Eq. 8); (c) number of nearest neighbors $K$ used in the KNN distance—top: for OOD scoring, bottom: for the sampling process.

regions—such as the comb, feathers, and claws for hens—which are distorted or abstracted, reducing resemblance to typical ID images while preserving overall class-like structure. This suggests that gradient-based guidance focuses on classifier-critical regions, producing samples near the decision boundary rather than only perturbing semantic description, as prior methods often do.

**Key Components in Our Method.** We further analyze the contribution of each component in GOOD through a controlled ablation study, summarized in Table 3. Rows (2)–(3) evaluate the effect of using only one type of guidance during OOD synthesis. Specifically, Row (2) uses image-level guidance (GOOD_img) alone, and Row (3) uses feature-level guidance (GOOD_feat) alone. Both variants employ balanced sampling across different step sizes to ensure diverse OOD generation, but they rely solely on the default energy score for detection. Row (4) represents the full GOOD pipeline *without* the unified OOD score, i.e., both image- and feature-level guidance are used for OOD synthesis while the energy score alone ($a = 0$ in Fig. 6(b)) is used for detection—this configuration is comparable to prior works such as DreamOOD and BOOD. Finally, Row (5) shows the complete GOOD framework, where both synthesis and detection jointly leverage image- and feature-level signals through the unified OOD score.

As shown in Table 3, removing either guidance mechanism significantly degrades performance (FPR95 rises from 17.70 to over 32), although each still surpasses the baseline ID classifier. Combining both guidance types improves detection by complementing pixel-level and feature-level cues, and balanced sampling further stabilizes performance. The unified OOD score provides the largest gain—integrating both signals yields the highest AUROC (96.28) and the lowest FPR95 (17.70).

Table 3: Ablation Study of the Key Components.

|     | Method | FPR95↓ | AUROC↑ | ID Acc↑ |
|-----|--------|--------|--------|---------|
| (1) | ID Classifier | 48.68 | 89.46 | 87.64 |
| (2) | w/o GOOD_feat | 32.23 | 92.29 | 86.98 |
| (3) | w/o GOOD_img | 32.93 | 91.98 | 87.24 |
| (4) | Balanced Sampling | 30.82 | 92.81 | 87.16 |
| (5) | Unified OOD Score | **17.70** | **96.28** | 87.16 |

**Ablation on the regularization weight $\lambda$.** As shown in Figure 6 (a), increasing $\lambda$ enhances OOD detection performance, but also leads to a gradual decline in ID accuracy due to stronger regularization. To balance this trade-off between OOD sensitivity and ID classification, we choose $\lambda = 2.5$, which achieves a solid AUROC of 96.28 while maintaining a competitive ID accuracy of 87.16.

**The coefficients of Unified OOD Score.** Figure 6 (b) illustrates the impact of the interpolation coefficient $w$ in Figure 6(b), which balances image-level and feature-level OOD scores. Adjusting $w$ allows us to modulate the influence of each component. Our results demonstrate that adaptively setting $w$ based on the KL divergence between the test and ID distributions significantly improves the robustness of GOOD across diverse OOD scenarios.

**Ablation on k in calculating k-NN distance.** We examine the impact of the number of nearest neighbors $k$ used in feature-level OOD scoring. As shown in Figure 6 (c), varying $k$ from 50 to 500 has minimal effect on OOD detection performance, with AUROC remaining consistently around 96.2. We further analyze the influence of $k$ on the sampling results of GOOD$_{feat}$. For each $k$, we generate 3,500 OOD samples using our balanced step-size strategy. The resulting energy distributions are highly similar, indicating that the choice of $k$ has limited impact under our sampling scheme.

# 5  Conclusion

In this paper, we propose GOOD, a training-free framework that leverages off-the-shelf classifiers to guide diffusion models for synthesizing informative and diverse out-of-distribution (OOD) samples. By introducing dual-level guidance—image-level based on pixel-space likelihood and feature-level derived from latent-space sparsity—GOOD avoids the instability of embedding perturbations and enables controllable sampling toward low-density, semantically meaningful OOD regions. We further propose a unified OOD score that integrates both types of guidance for robust detection. Extensive experiments demonstrate that GOOD significantly improves OOD detection performance across standard benchmarks. Beyond methodological gains, our work underscores a broader insight: generative models, when steered by discriminative knowledge, offer a powerful and scalable avenue for enhancing model reliability under distributional shift.

**Limitations and Future Work.** GOOD relies on the distribution learned by pre-trained diffusion models, which may limit the diversity of generated outliers. Future work will explore deeper integration of diffusion features with OOD objectives, enabling joint guidance in both pixel and feature spaces for more targeted and controllable outlier synthesis.

## Acknowledgement

This work was supported in part by the National Natural Science Foundation of China under Grant No. 62502200, the Jiangsu Provincial Science and Technology Major Project under Grant No. BG2024042, and the Natural Science Foundation of Jiangsu Province under Grant No. BK20251203. It was also supported by the Ministry of Education, Singapore, under its MOE AcRF Tier 2 programs (MOE-T2EP20221-0012 and MOE-T2EP20223-0002). Additional support was provided by the Nanjing Kunpeng & Ascend Center of Cultivation.

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

# A Appendix

## A.1 Related Work (Recap and Expansion)

### A.1.1 Out-of-distribution (OOD) Detection

OOD robustness is also critical for downstream applications [20, 42, 68, 39, 41, 40, 45]. Many early OOD detection methods rely on post-hoc processing of classifier outputs, such as logits [24, 23, 43, 47], model features [38, 51, 54, 61], or gradients [5, 31]. Other approaches directly modify classifier training by employing adjusted loss functions such as energy-based objectives [46], confidence-based rejection [11, 30, 37], or auxiliary self-supervised tasks [3, 65]. Recently, Outlier Exposure (OE) has significantly improved performance by leveraging large auxiliary datasets as pseudo-OOD samples, shaping conservative and robust decision boundaries; however, its effectiveness is limited by the availability and relevance of external datasets, thus motivating synthesized outlier methods. Approaches like VOS [16] and NPOS [62] generate synthetic outliers in feature space, while Dream-OOD [15] introduces a two-stage diffusion-based method that aligns and perturbs embeddings in Stable Diffusion's text-conditional space to produce pixel-level OOD images. Subsequent works such as BOOD [44] and NCIS [13] adopt this framework but employ adversarial perturbations and learned nonlinear invariants, respectively. In contrast, our method eliminates embedding alignment by directly guiding the diffusion sampling trajectory toward OOD regions via gradient-based pixel-level adjustments. This single-stage strategy significantly improves efficiency, robustness, and the diversity of synthesized OOD images.

### A.1.2 Guided Sampling with Diffusion Models

Generative approaches such as variational autoencoders (VAEs) and diffusion models have become fundamental tools for probabilistic data modeling [27, 18]. Diffusion models generate samples through iterative denoising, effectively estimating data gradients [57, 55], and naturally support post-hoc conditioning via optimization signals. Classifier-based guidance [58, 12] uses a noise-conditional classifier to provide gradients during sampling, while classifier-free guidance (CFG) [28] avoids extra classifiers by interpolating between conditional and unconditional predictions. However, both approaches require task-specific training, which limits scalability. In contrast, training-free guidance (TFG) guides sampling using differentiable target functions—such as classifiers or loss—without additional training. While several TFG methods have been proposed [22, 56, 8, 4, 73], most remain task-specific and lack unified theoretical foundations. Recently, Ye et al. [71] formalized a general TFG framework to unify such strategies. Building on this, our work extends training-free guidance to OOD samples generation by defining a target function that captures OOD signals via pre-trained classifiers and guiding the diffusion process to synthesize informative outliers.

## A.2 Theoretical Analysis

### A.2.1 Integration of GOOD into the TFG Framework

The key contribution of the paper *"TFG: Unified Training-Free Guidance for Diffusion Models"* [71] is the proposal of a unified and extensible algorithmic framework that enables conditional generation from unconditional diffusion models using off-the-shelf, differentiable target functions $f(x)$—without requiring any retraining. TFG generalizes and unifies several prior training-free sampling methods under a common formalism.

At each denoising step $t$, the diffusion model estimates the clean signal as:

$$x_{0|t} = \frac{x_t - \sqrt{1 - \bar{\alpha}_t} \cdot \epsilon_\theta(x_t, t)}{\sqrt{\bar{\alpha}_t}}.$$

To guide sampling, TFG applies gradients of a smoothed objective function:

$$\tilde{f}(x) = \mathbb{E}_{\delta \sim \mathcal{N}(0,I)} \left[ f \left( x + \bar{\gamma} \sqrt{1 - \bar{\alpha}_t} \cdot \delta \right) \right],$$

which stabilizes gradients by smoothing the predictor landscape via Gaussian convolution.

Guidance is introduced via two complementary strategies:

- **Variance guidance**, a second-order signal, uses gradients with respect to the noisy sample $x_t$:

$$\Delta_t = \rho_t \cdot \nabla_{x_t} \log \tilde{f}(x_{0|t}),$$

leveraging the covariance between $x_t$ and $x_{0|t}$ to adjust the sampling trajectory.

- **Mean guidance**, a first-order signal, applies gradients with respect to the predicted clean sample $x_{0|t}$:

$$\Delta_0 = \sum_{i=1}^{N_{\text{iter}}} \mu_t \cdot \nabla_{x_{0|t}} \log \tilde{f}(x_{0|t} + \Delta_0),$$

steering samples directly in data space toward higher scoring regions.

Moreover, TFG allows **Recurrence**—repeated application of guidance and denoising—to further refine sampling over $N_{\text{recur}}$ cycles, improving convergence and robustness to local optima.

This design space is formalized as:

$$\mathcal{H}_{\text{TFG}} = \left\{ (N_{\text{recur}}, N_{\text{iter}}, \bar{\gamma}, \rho, \mu) \right\},$$

which subsumes prior works such as DPS [8], LGD [56], MPGD [22], FreeDoM [73], and UGD [4] as special cases, enabling unified theoretical analysis and practical design.

**Extension to Our Method.** *Our method GOOD (Guided OOD sampling) fits naturally into the TFG framework by instantiating its guidance procedure with two novel, task-driven target predictors $f(x)$ tailored for out-of-distribution sample generation.*

**Target Predictor:** In line with the TFG formulation, we define a differentiable target function $f_c(x) : \mathcal{X} \to \mathbb{R}_+ \cup \{0\}$ that evaluates how well a sample $x$ aligns with an OOD objective conditioned on $c$. Following the conditional sampling framework:

$$p_0(x \mid c) = \frac{p_0(x) f_c(x)}{\int_{\tilde{x}} p_0(\tilde{x}) f_c(\tilde{x}) d\tilde{x}},$$

we seek to guide diffusion trajectories toward low-likelihood, feature-sparse regions representing diverse and informative OOD samples.

Inspired by post-hoc OOD scoring methods [24, 23, 43, 47, 61], we propose two concrete instantiations of $f_c(x)$:

- **Image-Level Predictor (GOOD$_{\text{img}}$)**: Based on the *free energy* [46] of a pretrained classifier $f$, we define

$$G_{ood}^{img}(x) := exp(E(x; f)) = \sum_{k=1}^{C} \exp(f_k(x)),$$

which approximates the negative log-likelihood of $x$ under the classifier's predictive distribution. Its gradient $\nabla_x \mathcal{E}(x; f)$ pushes samples from high-density ID regions to low-density OOD regions in pixel space.

- **Feature-Level Predictor (GOOD$_{\text{feat}}$)**: To capture structural sparsity, we compute the $k$-nearest-neighbor distance in the normalized feature space:

$$G_{ood}^{feat}(x) := D_k(x; f) = \|z(x) - z^{(k)}\|_2, \quad z(x) = \frac{f^{(1:L)}(x)}{\|f^{(1:L)}(x)\|_2},$$

where $z^{(k)}$ is the $k$-th nearest feature vector from an in-distribution memory bank. Gradients of this score encourage exploration of semantically novel and underrepresented directions in feature space.

These target predictors define complementary guidance signals and are differentiable, enabling their seamless integration into the TFG framework.

**Instantiation within TFG.** We instantiate GOOD using the following configurations in TFG:

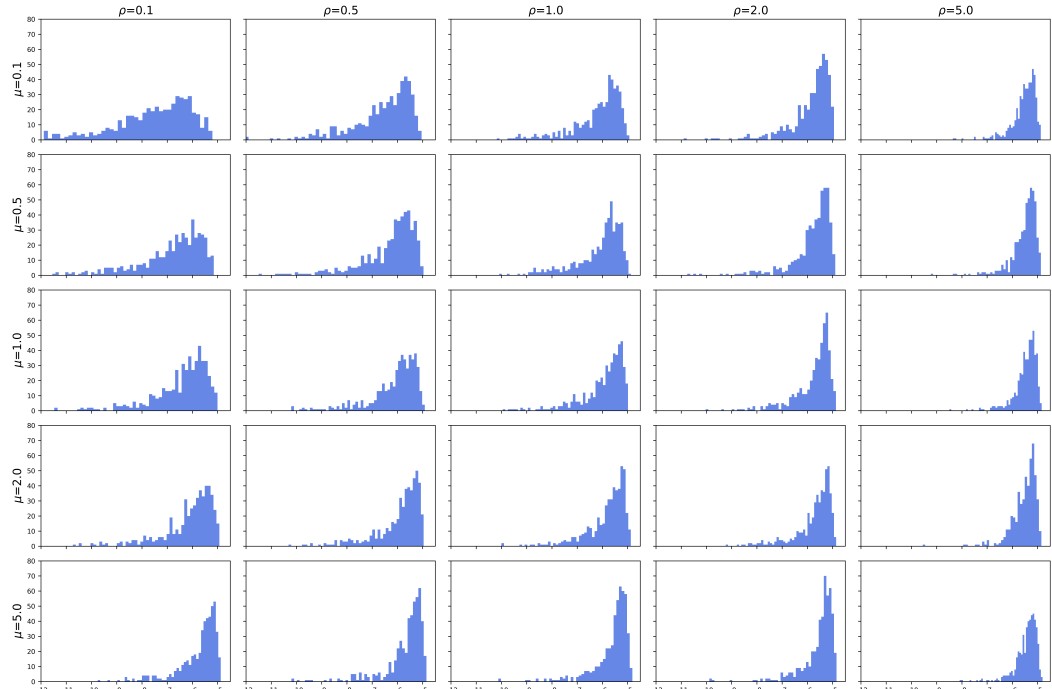

Figure 7: Energy distributions of 500 OOD samples generated under each $(\bar{\rho}, \bar{\mu})$ configuration on ImageNet ($\bar{\gamma} = 0.1$). Each subplot shows the histogram of negative energy scores (i.e., $\log \sum_k \exp f_k(x)$) under a fixed pair of variance guidance strength $\bar{\rho}$ (columns) and mean guidance strength $\bar{\mu}$ (rows). Higher $\bar{\rho}$ and $\bar{\mu}$ tend to produce lower-energy (more outlying) samples, while lower values yield samples closer to the in-distribution manifold. This grid highlights the tradeoff between sample extremeness and diversity, supporting our use of balanced sampling across parameter combinations.

- **(a) Implicit Dynamic:** Gaussian smoothing is applied to $f$ via

$$\tilde{G}_{ood}(x) = \mathbb{E}_{\delta \sim \mathcal{N}(0,I)}[G_{ood}(x + \bar{\gamma}\sqrt{1 - \bar{\alpha}_t}\delta)]$$

  to ensure smooth and stable gradient fields;

- **(b) Variance Guidance:** Guidance on the noisy sample $x_t$ via second-order signal $\nabla_{x_t} \log \tilde{G}_{ood}(x_{0|t})$;

- **(c) Mean Guidance:** Guidance on the denoised estimate $x_{0|t}$ via first-order optimization $\nabla_{x_{0|t}} \log \tilde{G}_{ood}(x_{0|t})$.

For computational efficiency, we omit recurrence and set $N_{\text{recur}} = N_{\text{iter}} = 1$, since outlier training typically requires generating a large number of samples, and iterative guidance would significantly slow down the sampling process.

By plugging our OOD-specific target predictors into TFG's modular framework, GOOD enables principled, training-free guidance of diffusion sampling toward low-density and classifier-sensitive regions. This design allows us to generate boundary-adjacent OOD samples that are diverse, semantically aligned, and highly effective for outlier exposure training.

### A.2.2 Hyperparameter Selection

The TFG framework introduces a structured design space:

$$\mathcal{H}_{\text{TFG}} = \left\{ (N_{\text{recur}}, N_{\text{iter}}, \bar{\gamma}, \rho, \mu) \right\},$$

where each hyperparameter governs a specific guidance behavior. We now detail our choices for each component in the context of OOD sampling, balancing generation quality, diversity, and efficiency.

**Recurrence ($N_{\text{recur}}$) and Mean Iteration ($N_{\text{iter}}$).** Recurrence amplifies the guidance signal by repeating the guidance–denoise–reinject cycle, while $N_{\text{iter}}$ controls the number of inner steps used to refine the clean estimate $x_{0|t}$. Although these operations can improve sample quality, they are computationally expensive. Since outlier exposure training typically requires synthesizing thousands of OOD samples, we prioritize efficiency and fix both to one step: $N_{\text{recur}} = N_{\text{iter}} = 1$.

**Smoothing Strength ($\bar{\gamma}$).** The Gaussian smoothing in implicit dynamics stabilizes the gradient field of $f(x)$, especially in low-density regions, by averaging local variations. We select $\bar{\gamma}$ based on resolution: 0.1 for low-resolution CIFAR and 0.001 for high-resolution ImageNet. These values were determined via a lightweight beam search focused on the energy distribution of generated images.

**Variance Guidance Strength ($\rho$).** The coefficient $\rho_t$ scales the gradient $\nabla_{x_t} \log \tilde{f}(x_{0|t})$ applied in the noisy space. As variance guidance encodes second-order information, it is particularly useful during early diffusion steps when $x_t$ is heavily corrupted and carries limited semantic content.

Following the design in the original TFG paper [71], we adopt an increasing time-dependent schedule:

$$\rho_t = \bar{\rho} \cdot \frac{\alpha_t}{\sum_{s=1}^{T} \alpha_s},$$

where $\bar{\rho}$ controls the overall strength of variance guidance. This increasing structure has been empirically validated across multiple tasks—including Gaussian deblurring, super-resolution, label guidance, and style transfer—demonstrating its effectiveness in gradually shifting the focus from low-level pixel alignment to higher-level structural refinement as the denoising process unfolds.

**Mean Guidance Strength ($\mu$).** Similarly, mean guidance operates on the denoised estimate $x_{0|t}$ and becomes more impactful in later steps, when the sample becomes semantically clearer and closer to the data manifold. We therefore adopt the same increasing schedule:

$$\mu_t = \bar{\mu} \cdot \frac{\alpha_t}{\sum_{s=1}^{T} \alpha_s}.$$

This schedule ensures that mean guidance remains weak during early steps, where gradients are noisy, and grows stronger as the sample becomes more refined.

**Sampling with Diverse Parameter Combinations Rather than Single Optimal.** Instead of searching for a single optimal $(\bar{\rho}, \bar{\mu})$ pair, we adopt a balanced sampling strategy using a fixed grid of parameter combinations. Specifically, we sample from the Cartesian product $\bar{\rho}, \bar{\mu} \in \{0.1, 0.5, 1.0, 2.0, 5.0\}$, resulting in 25 configurations. Each configuration yields OOD samples with different anomaly intensities and semantic characteristics. Figure 7 visualizes the negative energy distributions of 500 samples per configuration on ImageNet. As $\bar{\rho}$ and $\bar{\mu}$ increase (from left to right and top to bottom), the generated samples shift toward lower energy regions, indicative of stronger outlier characteristics. Conversely, low guidance strengths produce samples closer to the in-distribution manifold. This tradeoff between semantic plausibility and anomaly severity is crucial: overly strong guidance may yield unrealistic or trivial outliers, while weak guidance may fail to leave the data manifold meaningfully.

By covering the full grid, our approach generates a continuum of near-OOD samples spanning subtle to extreme shifts. This diversity enhances outlier exposure training by reducing overfitting to a narrow anomaly profile and promoting robustness to real-world distributional shifts.

## A.3 Implementation Details and Datasets

### A.3.1 Details of Balanced Sampling

To support diverse and robust outlier exposure, we generate a wide spectrum of OOD samples using a balanced grid-based strategy described earlier. Specifically, we sample a large number of images across different guidance strengths, covering both image-level and feature-level scores. The sampling setup is consistent across both ImageNet-100 and CIFAR-100.

Table 4: Comparison of computational stages required by existing outlier exposure methods. Traditional approaches involve either expensive manual collection or multi-stage generation pipelines, while GOOD eliminates most stages by directly guiding diffusion sampling.

| Method | Manual Data Collection | ID Embedding Alignment | OOD Embedding Sampling | OOD Sample Generation | Training and Detection |
|---|---|---|---|---|---|
| WOODS [33] | ✓ | | | | ✓ |
| SAL [14] | ✓ | | | | ✓ |
| Dream-OOD [15] | | ✓ | ✓ | ✓ | ✓ |
| FodFoM [7] | | ✓ | ✓ | ✓ | ✓ |
| NCIS [13] | | ✓ | ✓ | ✓ | ✓ |
| BOOD [44] | | ✓ | ✓ | ✓ | ✓ |
| **GOOD (Ours)** | | | | ✓ | ✓ |

**Image-level guidance (GOOD$_{img}$).** For image-based guidance, we adopt a full Cartesian product of guidance strengths with $\bar{\rho}, \bar{\mu} \in \{0.1, 0.5, 1.0, 2.0, 5.0\}$, resulting in 25 unique parameter pairs. For each parameter configuration, we sample 5 images per class across 100 classes, yielding: $25 \times 5 \times 100 = \mathbf{12{,}500}$ samples.

**Feature-level guidance (GOOD$_{feat}$).** For feature-based guidance, we follow the kNN-based sparsity scoring method proposed in [61]. Unlike energy scores, kNN distances are not normalized to $[0, 1]$ due to their small dynamic range; instead, we apply a fixed scaling factor of 5 before computing gradients. For simplicity, we set $\bar{\rho} = \bar{\mu}$ and choose 7 representative values: $\{0.2, 0.4, 1.0, 1.5, 2.0, 3.0, 4.0\}$. For each configuration, we generate 15 images per class, resulting in: $7 \times 15 \times 100 = \mathbf{10{,}500}$ samples.

This balanced sampling strategy ensures that our generated OOD dataset covers a rich diversity of anomaly levels—ranging from near-distribution hard negatives to semantically abstract outliers—thereby improving generalization during training and evaluation.

### A.3.2 Computational Cost Comparison

Outlier exposure methods generally fall into two categories: data-collection-based and generation-based. The former, such as WOODS [33] and SAL [14], rely on manually curated external datasets, which are costly to collect and often domain-dependent. The latter—including Dream-OOD [15], FodFoM [7], NCIS [13], and BOOD [44]—adopt multi-stage generation pipelines that incur substantial computational overhead. These generation-based approaches typically involve: (1) constructing or aligning latent feature spaces from ID embeddings, (2) synthesizing OOD features within these spaces, (3) decoding them into images using pretrained or fine-tuned generative models (e.g., diffusion or GANs), and (4) training a detection model on the resulting samples. As summarized in Table 4, all stages incur non-trivial computational and engineering costs.

In particular, Dream-OOD reports a total compute time exceeding 16 hours on ImageNet-100, including 8.2h for latent space construction, 10.1h for diffusion-based image generation, and 8.5h for training. BOOD reports similar costs, with 0.62h for building the latent space, 0.1h for OOD feature synthesis, 7.5h for image generation, and 8.5h for regularized training. NCIS requires up to 13h for ID embedding extraction alone (on 8×A100 GPUs), plus additional time for training the cVPN projection network.

In contrast, our method GOOD simplifies the pipeline by directly guiding the diffusion sampling process with gradients from a pretrained classifier. As shown in the shaded row of Table 4, GOOD bypasses most stages: it does not require latent space construction, embedding alignment, or feature synthesis. The only computational components are the forward/backward passes of the classifier during sampling, and the final outlier exposure training. In practice, each $512 \times 512$ image generated with OOD guidance (based on a $224 \times 224$ classifier input) takes approximately $5.7\times$ longer than unconditional sampling. However, unlike prior methods that typically require generating over 100,000 OOD samples, our method only needs about one-fifth as many—making the total computational cost comparable or even lower. As a result, the overall computational cost remains manageable. This efficient design supports scalable, high-quality OOD sample generation and enables seamless integration into large-scale training pipelines without retraining or architectural changes.

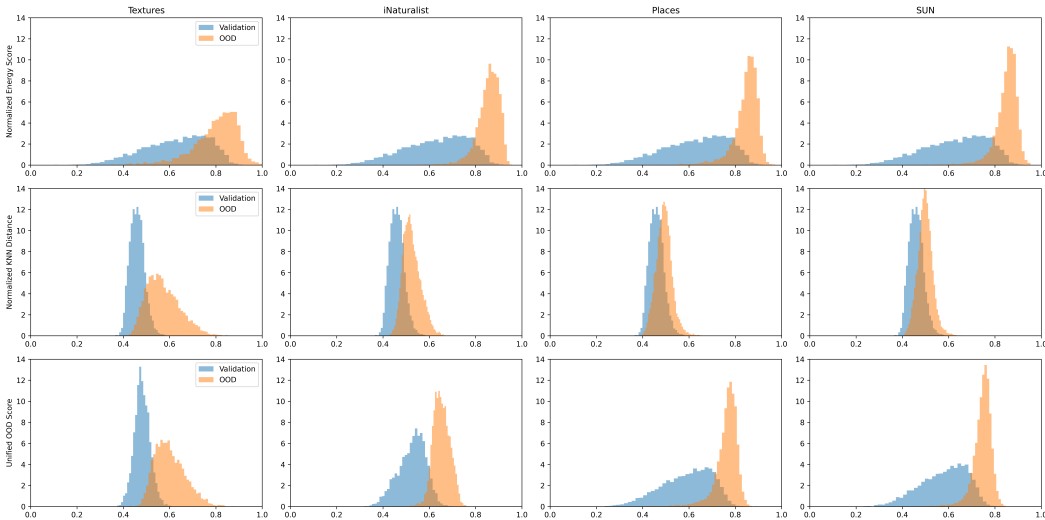

Figure 8: Distributions of OOD detection scores across four OOD datasets (Textures, iNaturalist, Places, and SUN) and three score types: energy-based score (top row), k-nearest neighbor (KNN) distance (middle row), and a unified OOD score (bottom row) that adaptively fuses the two based on KL divergence. Each subplot compares the score distributions for the ID validation set (Imagenet-100) and the corresponding OOD dataset. All scores are normalized to the range [0, 1].

### A.3.3 Distribution Analysis across OOD Datasets

For ImageNet-100, we follow [32] and evaluate OOD detection performance on four standard benchmarks: Textures [9], iNaturalist [63], Places [74], and SUN [66]. For CIFAR-100, we use five common OOD datasets: SVHN [49], Places365 [74], LSUN-R [72], ISUN [67], and Textures [9].

To understand how different scoring functions separate ID and OOD samples, Figure 8 presents histograms of normalized detection scores across the four ImageNet-100 OOD datasets. Each column corresponds to a different OOD dataset, and each row shows a different score type: energy-based score (top), k-nearest neighbor (KNN) distance in feature space (middle), and our proposed unified score (bottom). We observe that the energy score distributions (top row), which reflect low image likelihood, tend to have broader spread and more overlap with ID validation scores—especially for iNaturalist, Places, and SUN. In contrast, the KNN-based feature distance (middle row) produces sharply peaked and better-separated distributions for certain datasets, particularly Textures, indicating that these OOD samples are more distinguishable in feature space than in pixel space.

Interestingly, the separability patterns differ by dataset: Textures is clearly more separable via feature distance, while the other three datasets (iNaturalist, Places, and SUN) are more distinguishable by energy score. To leverage the strengths of both signals, we compute a KL divergence between ID and OOD score distributions for each type and use it to adaptively weight the two, producing a unified OOD score (bottom row). As shown in the last row of Figure 8, this fused score consistently improves separation across all four datasets by emphasizing the more discriminative signal for each case.

This analysis supports the use of our unified score as a reliable, adaptive OOD detection signal that balances image-space and feature-space cues.

### A.4 Additional Results and Case Studies

### A.4.1 Additional Results on CIFAR

To verify that our guidance directions reflect semantically meaningful trajectories relative to the classifier, we conduct two controlled experiments on CIFAR datasets, each designed to probe the behavior of GOOD in feature space.

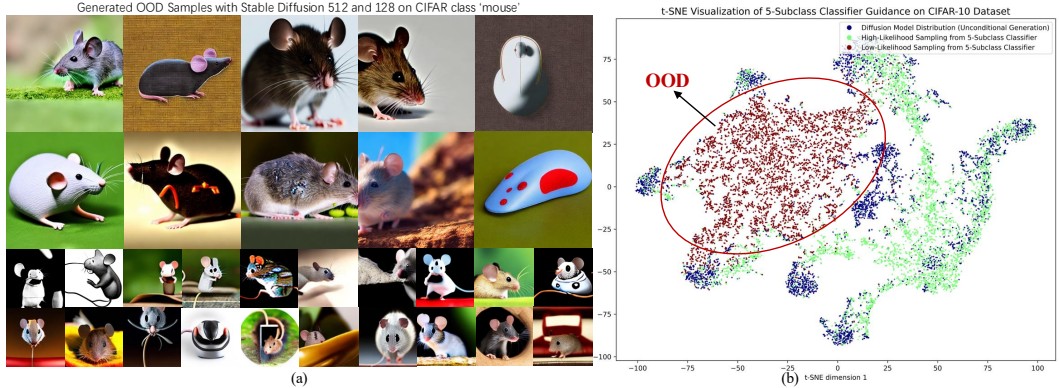

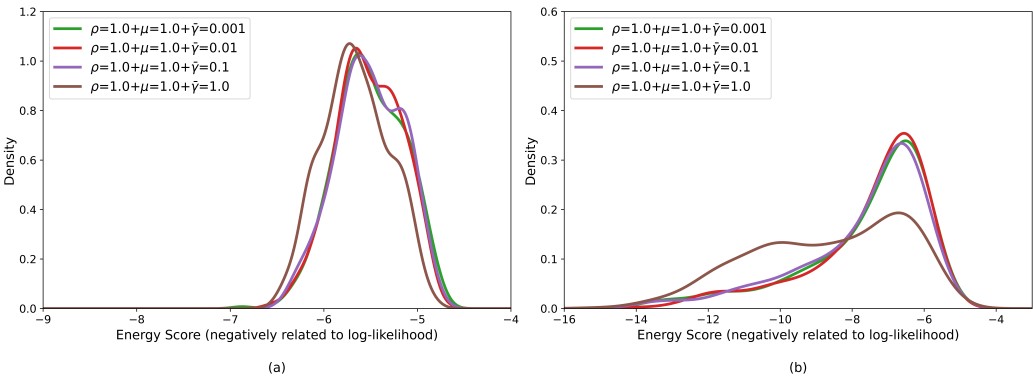

Figure 9: (a) OOD samples from the CIFAR-100 mouse class using Stable Diffusion (512/128). (b) t-SNE plot of samples from a 5-class CIFAR-10 classifier using DDPM: OOD guidance pushes samples toward unseen classes; ID guidance pulls them into known class regions.

Figure 10: Energy score distributions of OOD samples generated with different $\bar{\gamma}$ values, under fixed $\bar{\rho} = \bar{\mu} = 1.0$. Each curve is computed from 500 samples. (a) ImageNet-100, (b) CIFAR-100.

**(a) OOD sample diversity via Stable Diffusion.** We first generate OOD samples conditioned on the CIFAR-100 class "mouse" using a high-resolution Stable Diffusion model (512/128). Figure 9(a) showcases a diverse range of mouse-related outputs. The samples vary from realistic mice to abstract renderings, cartoons, and even symbolic representations—demonstrating how GOOD naturally induces both semantic preservation and controlled abstraction. This supports its ability to generate meaningful OOD variants within the scope of a class concept.

**(b) Semantic alignment in latent space via t-SNE.** Next, we study the semantic effect of guidance using a $32 \times 32$ DDPM model trained on CIFAR-10. We partition the 10 classes into two halves: five seen classes are used to train a classifier, while the other five are held out. Using this setup, we perform three types of sampling: (1) unconditional generation, (2) ID-guided sampling (i.e., maximizing likelihood under the seen-class classifier), and (3) OOD-guided sampling (i.e., minimizing likelihood under the same classifier).

Figure 9(b) visualizes the resulting samples using t-SNE. Unconditional samples (green) spread across the generative manifold. ID-guided samples (blue) are pulled toward the classifier's known class regions, forming tight clusters. In contrast, OOD-guided samples (red) are pushed away from the seen-class subspace and drift toward unfamiliar, in-between regions. This contrast highlights that GOOD's guidance directions are indeed semantically aligned: positive guidance avoids known categories, while negative guidance reinforces them.

### A.4.2 Additional Ablation Study

**Effect of $\bar{\gamma}$ in TFG.** The hyperparameter $\bar{\gamma}$ controls the strength of Gaussian smoothing in the implicit dynamics of TFG. It determines how much noise is added when computing the smoothed target function $\tilde{f}(x)$, which in turn affects the stability and directionality of the guidance signal.

To evaluate its effect, we fix $\bar{\rho} = \bar{\mu} = 1.0$ and vary $\bar{\gamma} \in \{0.001, 0.01, 0.1, 1.0\}$, generating 500 OOD samples for each setting. Figure 10 shows the resulting energy score distributions on ImageNet-100 (left) and CIFAR-100 (right). These scores are inversely related to image likelihood, and are commonly used to characterize the degree of distributional shift.

We observe that small values of $\bar{\gamma}$ (e.g., 0.001, 0.01) produce sharp, unimodal distributions centered around a typical energy range, indicating stable and consistent guidance. As $\bar{\gamma}$ increases, the distributions become flatter or multimodal—especially on CIFAR—suggesting that excessive smoothing can inject noisy or less directional gradients, resulting in more diverse but potentially less semantically aligned samples.

Overall, GOOD exhibits robustness to a range of $\bar{\gamma}$ values. However, we find that moderate values (e.g., 0.1 for ImageNet, 0.001 for CIFAR) provide the best trade-off between stability and anomaly diversity. These values are therefore used as default in all main experiments.

## A.5 Additional Visualizations

To further illustrate the generative behavior of our approach, we present additional OOD samples produced by GOOD$_{\text{img}}$ and GOOD$_{\text{feat}}$ on ImageNet-100. For each of the first 72 classes, we generate one sample per class under each guidance strategy, using fixed parameters.

Figures 11, 13, and 15 show results from GOOD$_{\text{img}}$ with $\bar{\rho} = \bar{\mu} = 5$, where guidance is based on low image-level likelihood (free energy). These samples tend to exhibit a global shift toward abstraction—often distorting not just local texture but also the overall scene composition. In many cases, the image foreground, background, and object boundaries are all rendered with atypical patterns, surreal colors, or unnatural materials. Nonetheless, class semantics such as shape or context remain loosely preserved, resulting in globally outlying yet semantically grounded samples.

In contrast, Figures 12, 14, and 16 show results from GOOD$_{\text{feat}}$ with $\bar{\rho} = \bar{\mu} = 4$, where guidance targets sparsity in the deep feature space. Unlike GOOD$_{\text{img}}$, this variant preserves the overall image realism and structure but introduces rare or atypical class-specific features—such as uncommon textures, poses, or environmental contexts. For example, cats with distorted fur patterns or birds in unfamiliar postures. These samples lie closer to the data manifold but remain feature-wise distinctive, resembling "hard negatives" rather than globally abstract outliers.

Together, these visualizations highlight the complementary behavior of the two guidance modes: GOOD$_{\text{img}}$ encourages holistic abstraction, while GOOD$_{\text{feat}}$ promotes subtle novelty. This duality enables us to generate a diverse spectrum of OOD examples—ranging from boundary-adjacent to semantically twisted—without retraining, simply by switching the guidance target.

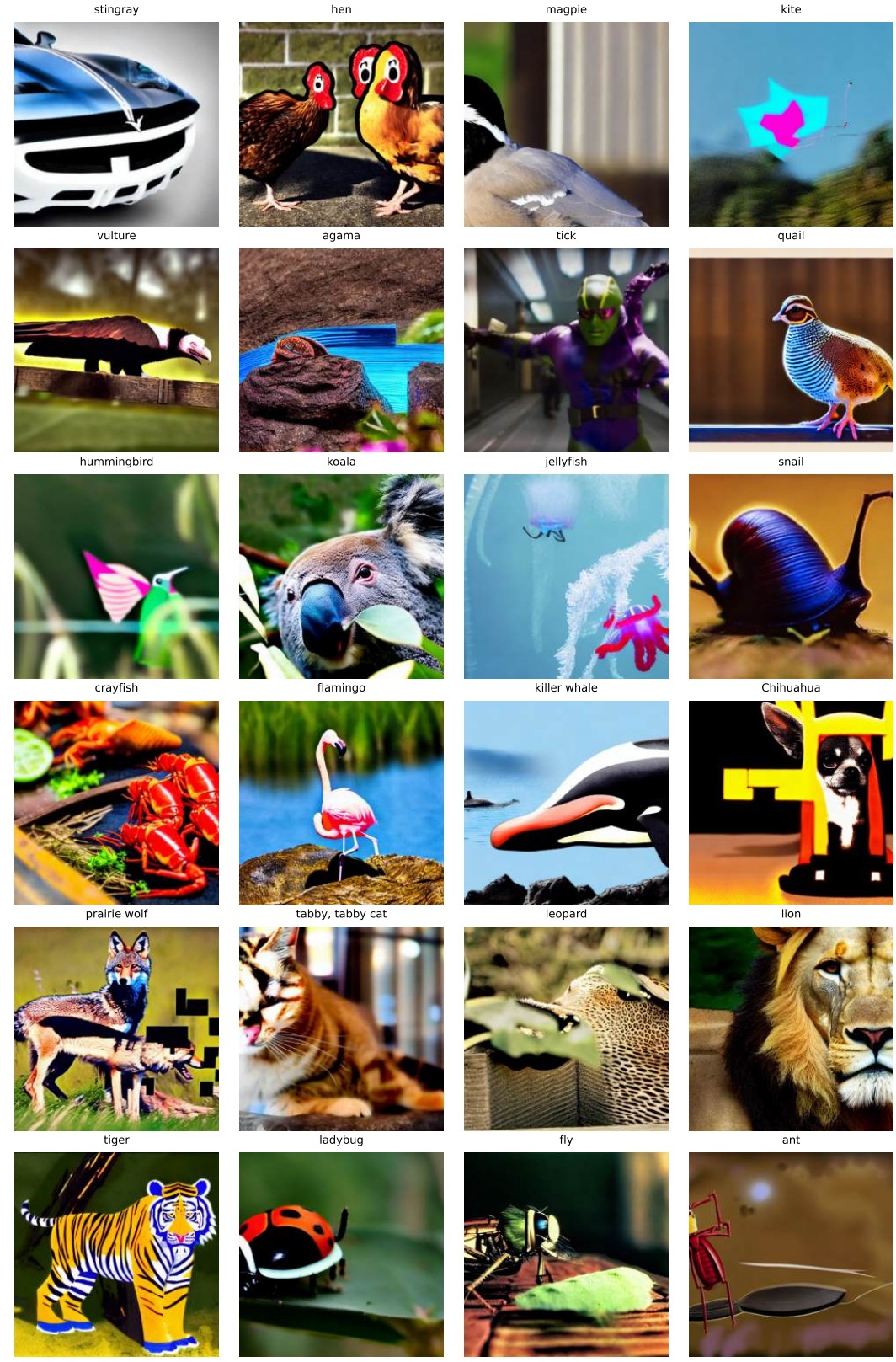

Figure 11: OOD samples generated by GOOD$_{img}$ on ImageNet, guided by low image likelihood (free energy). One sample per class is shown.

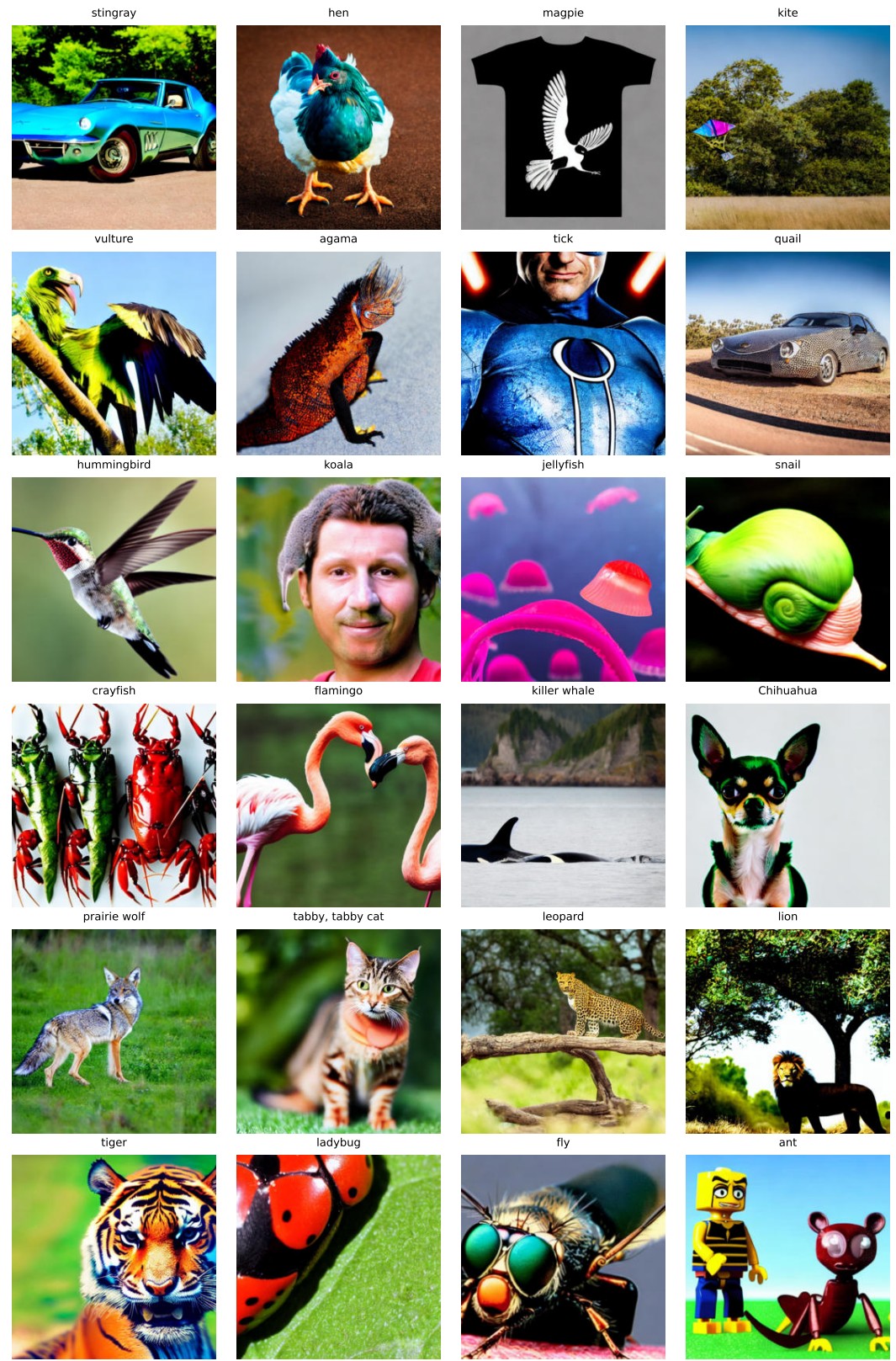

Figure 12: OOD samples generated by $\text{GOOD}_{\text{feat}}$ on ImageNet, guided by feature-space sparsity (kNN distance). One sample per class is shown.

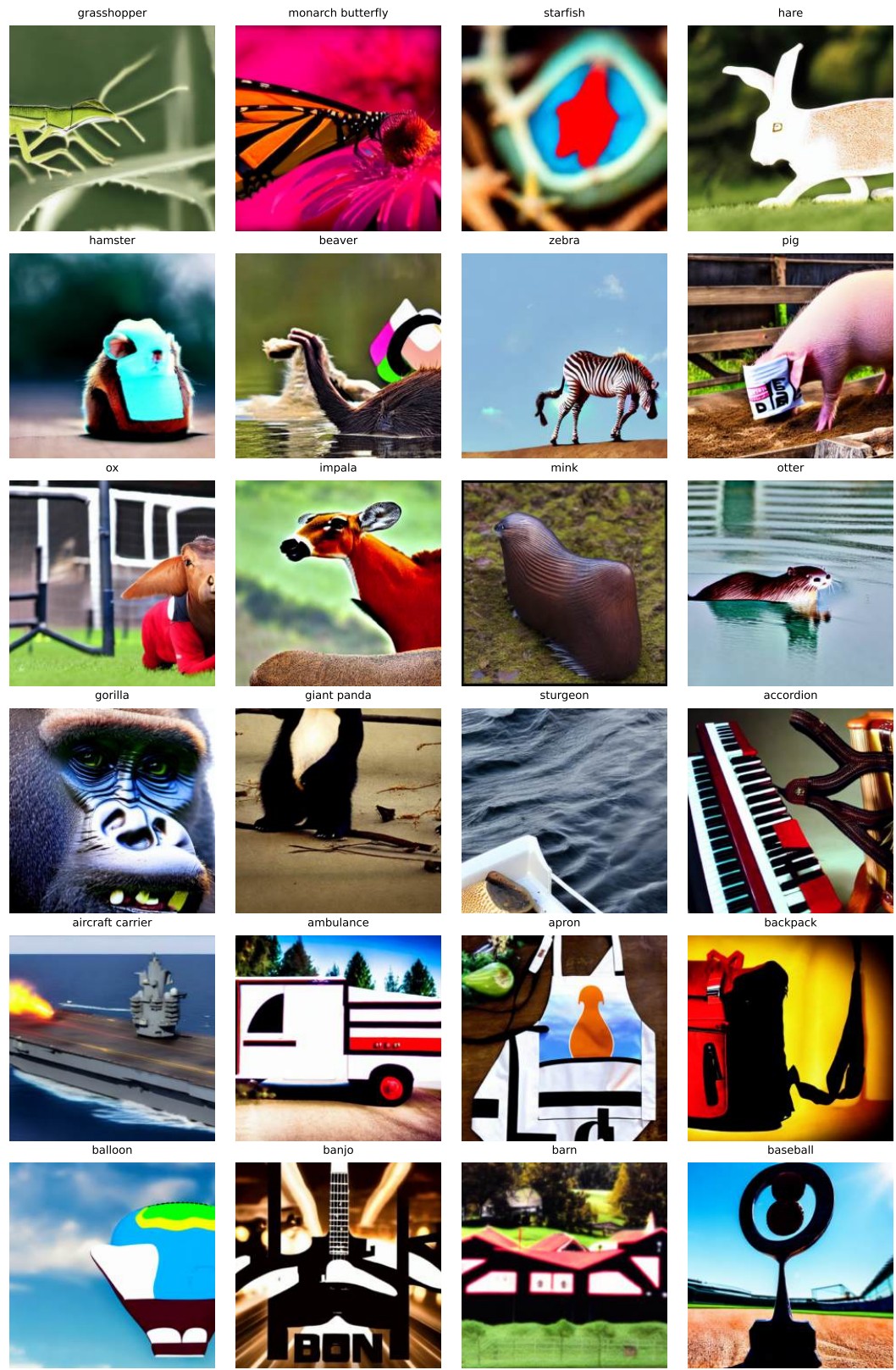

Figure 13: OOD samples generated by GOOD_img on ImageNet, guided by low image likelihood (free energy). One sample per class is shown.

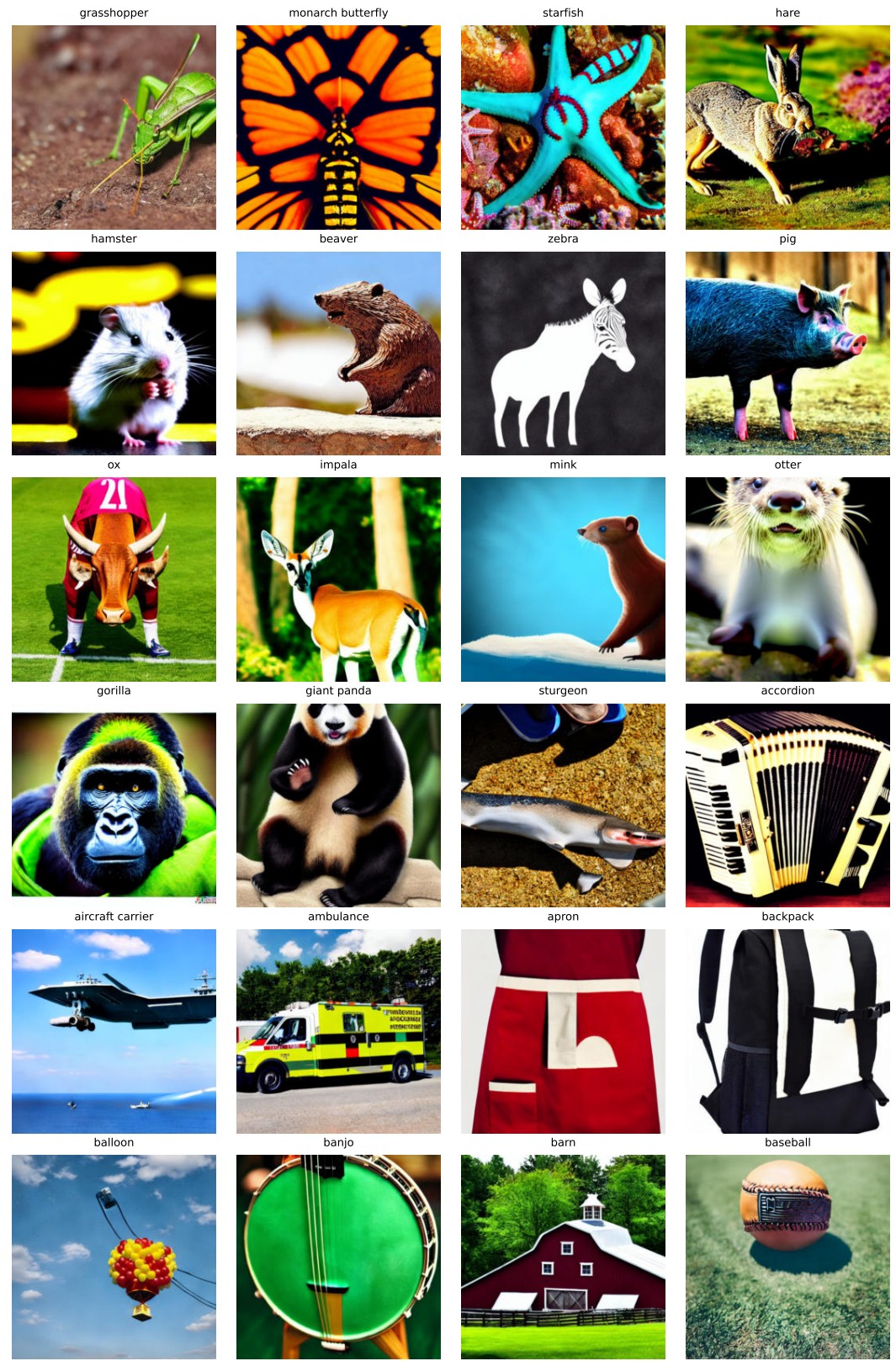

Figure 14: OOD samples generated by GOOD_feat on ImageNet, guided by feature-space sparsity (kNN distance). One sample per class is shown.

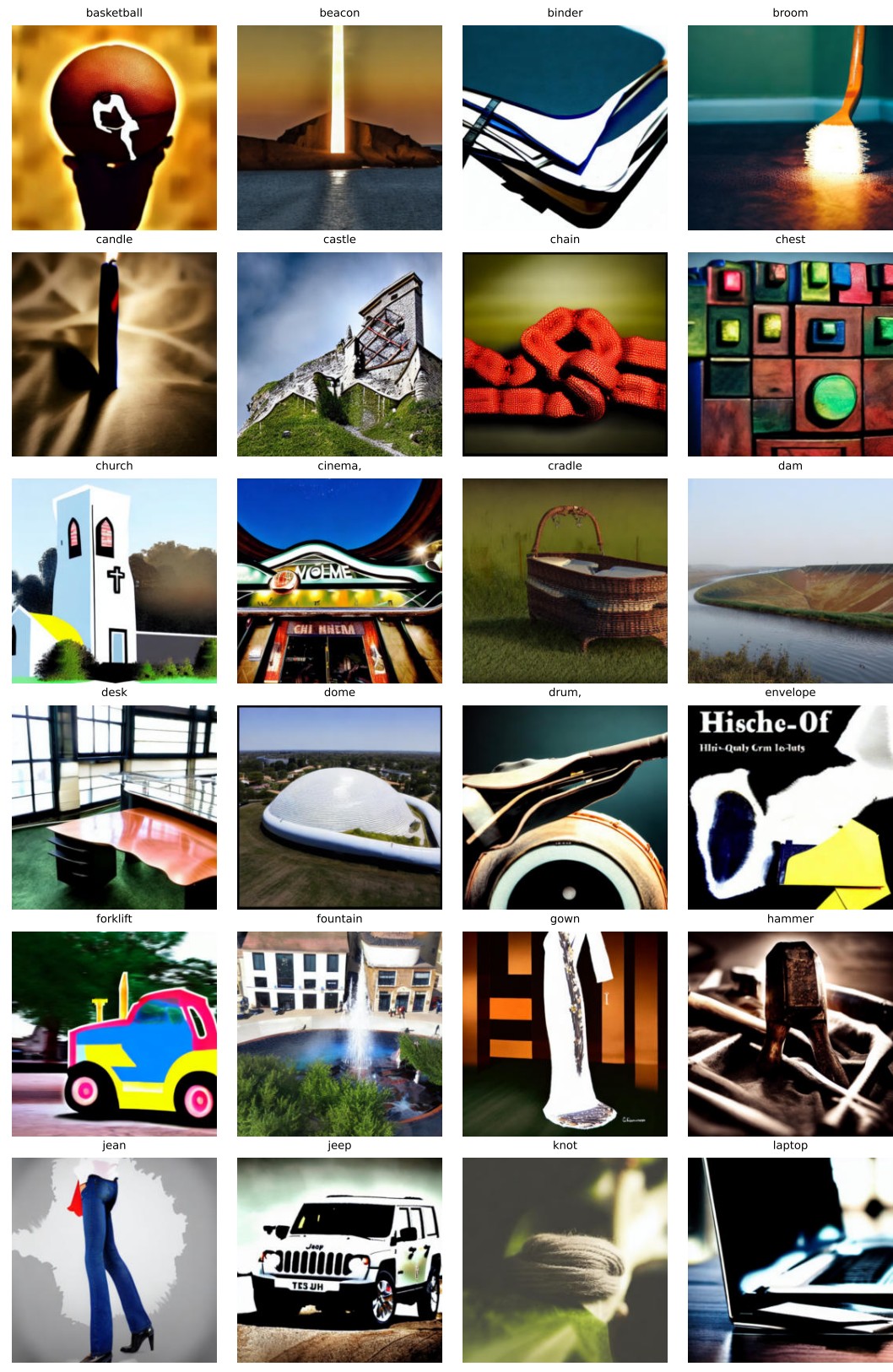

Figure 15: OOD samples generated by GOOD$_{\text{img}}$ on ImageNet, guided by low image likelihood (free energy). One sample per class is shown.

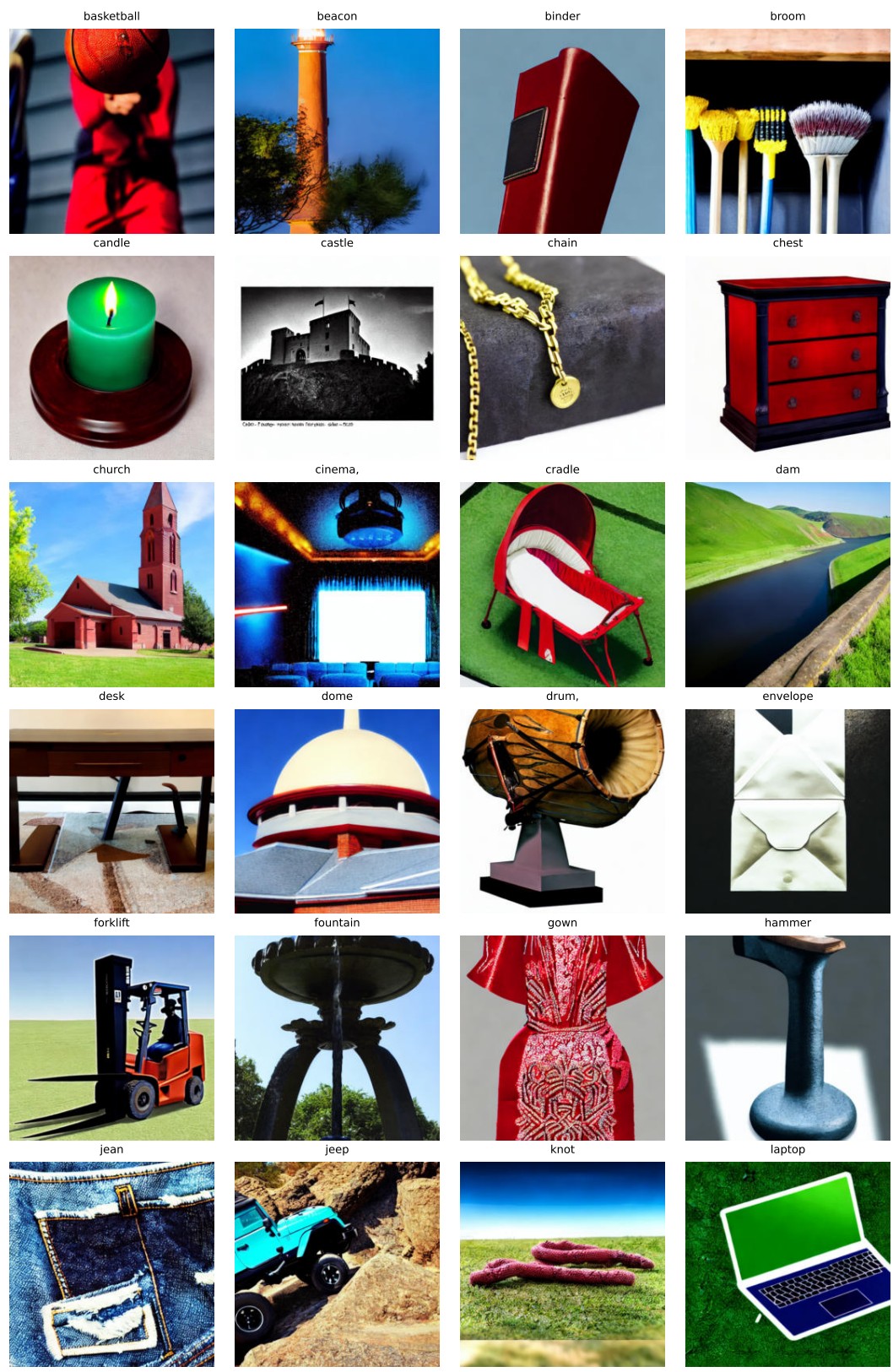

Figure 16: OOD samples generated by GOOD_feat on ImageNet, guided by feature-space sparsity (kNN distance). One sample per class is shown.

