# OpenReview forum: "GOOD: Training-Free Guided Diffusion Sampling for Out-of-Distribution Detection"
_NeurIPS.cc/2025/Conference — NeurIPS 2025 poster_

### Official Review · Reviewer_5MHc · 2025-06-30

**Clarity:** 2
**Significance:** 4
**Originality:** 3
**Rating:** 4
**Confidence:** 4

**Summary:**

This paper proposes a novel framework GOOD which harnesses diffusion models to generate OOD images to improve machine learning models' OOD detection performance. During the process of synthesizing images, GOOD combines image-level and feature-level guidances, forcing the diffusion model to sample towards OOD regions. Compared to previous methodologies, GOOD does not involve ID embedding alignment, which reduces computational costs. GOOD also achieve wonderful SOTA performance on ImageNet and CIFAR datasets.

**Questions:**

1. Could the authors clarify the potential reasons why GOOD shows comparatively worse performance on PLACES365 compared to FodFoM?
2. Could the authors provide the performance metrics of other OOD detection methods when integrated with the proposed unified OOD score?
3. Why generating 12500 GOOD<sub>img</sub> while generating 10500 GOOD<sub>feat</sub>? Is there a theoretical basis for this imbalance, and how does the quantity of synthesized OOD images impact overall performance?
4. I wonder how does the choice of ID classifier affect the performance of OOD detection?

**Ethical Concerns:**

["NO or VERY MINOR ethics concerns only"]

**Final Justification:**

After reading the rebuttal and other reviews, I agree that this paper has many weaknesses and decreases my rating.

**Limitations:**

Yes

**Quality:**

3

**Strengths And Weaknesses:**

Strengths:

    1. The paper provides various of high-quality images including framework and insightful visualizations.

    2. The authors include clear hyperparameters analysis and ablation studies, which help to understand the effectiveness of difference sections of the framework deeply. The experimental setup also provides detailed training parameters, making the reproduction easier.

    3. This methodology does not require embedding alignment, reducing computational cost in a novel way. The proposed method demonstrates superior performance compared to existing techniques on the evaluated datasets.

    4. The paper provides an in-depth analysis of OOD datasets, revealing the relationship between the model's performance and the distributions of OOD datasets.

Weaknesses:

    1. While the explanation of the TFG framework is provided in the appendix, its inclusion in the main body of the paper is essential for clarity. A brief mention in Section 3.2 and Algorithm 1 is insufficient, leaving the methodology unclear to the reader.

    2. The framework's reliance on a large number of hyperparameters may complicate the fine-tuning process, potentially hindering its practical application and adoption.

    3. In table 2, the dataset "PLACES365" has a typo.

---

> ### Author Rebuttal · Authors · 2025-07-31
>
> # Q1 Hyperparameter Sensitivity and Generalization
>
> (Weakness 2)
>
> **We emphasize that GOOD does not require extensive hyperparameter tuning to perform well.** The hyperparameters in our method include the guidance strength $\rho$ and $\mu$, the KNN distance parameter $k$, the regularization weight $\lambda$, and the coefficient $a$ for the unified OOD score.
>
> Among all hyperparameters, the guidance strengths $\rho$ and $\mu$ are the most challenging to tune, as they control how far the generated samples deviate from the ID manifold during diffusion. To address this, **rather than searching for a single optimal configuration, we adopt a balanced sampling strategy across a range of values (e.g., {0.1, 0.5, 1, 2, 5}), as detailed in Section 3.2**. This approach not only avoids tedious hyperparameter tuning, but also enables us to **construct a diverse OOD sample set** that spans multiple levels of anomaly severity—leading to improved robustness and coverage.
>
> The coefficient $a$ in the unified OOD score, which balances energy-based and kNN-based scores, has a noticeable impact on performance due to varying separability in OOD datasets. While tuning $a$ can improve performance, our method already performs well without it, and the unified score serves as a refinement, not a core dependency.
>
> For the remaining hyperparameters such as $k$ and $\lambda$, we conduct a detailed sensitivity analysis in Section 4.3 and Appendix A.2.2. Results show that the method is robust to a wide range of values, and default configurations suffice across all tested datasets.
>
> # Q2 Effect of OOD Sample Size
>
> (Question 3)
>
> Thank you for your insightful question regarding the number of **GOOD\_img** (12,500) and **GOOD\_feat** (10,500) generated samples. The discrepancy in sample sizes originated from the initial experimental process. Initially, we sampled 500 OOD images for each combination of guidance strengths across both types of guidance. After analyzing the distributions of these samples, we refined the parameter ranges and completed sampling.
>
> For **GOOD\_img**, we used 5 guidance strengths $\in \{0.1, 0.5, 1, 2, 5\}$, and for **GOOD\_feat**, 7 strengths $\in \{0.2, 0.4, 1, 1.5, 2, 3, 4\}$. **This imbalance was unintentional and arose from the exploratory phase, where we initially selected different quantities for each guidance type**. To prevent computational waste, we standardized the sample sizes to approximately 10,000 for both types.
>
> For consistency and methodological rigor, we now use the same set of guidance strengths {0.1, 0.5, 1, 2, 5} for both **GOOD\_img** and **GOOD\_feat**, resulting in equal sample sizes. The results for 20,000 samples (31.13 vs. 30.82 for **GOOD\_img** and 92.46 vs. 92.81 for **GOOD\_feat**) demonstrate that the slight difference in sample sizes has a negligible impact on performance. However, we also observed diminishing returns with additional generated samples, underscoring the trade-off between performance improvements and computational cost.
>
> Thus, we limited the generated samples to approximately 20,000, significantly reducing computational overhead compared to methods generating 100,000 samples. Our approach, which produces near-OOD samples closely aligned with ID data, ensures robust OOD detection with fewer samples, maintaining both efficiency and performance.
>
>
>
> # Q3 Impact of ID Classifier Choice
>
> (Question 4)
>
> Thank you for your insightful question regarding the impact of the ID classifier choice on OOD detection performance.
>
> Our experiments indicate that different backbone architectures exhibit varying OOD detection performances across datasets. For instance, Vision Transformers (ViTs) like ViT-B16 and convolutional networks such as ConvNeXt demonstrate notably better performance on the Textures dataset compared to ResNet34. **This discrepancy can be attributed to architectural differences: ViTs leverage self-attention mechanisms that capture long-range dependencies, while ConvNeXt integrates modern design elements inspired by transformers, enhancing feature extraction capabilities.**
>
> In contrast, ResNet34, a traditional convolutional neural network, may not capture fine-grained texture details as effectively, leading to relatively lower performance on texture-rich datasets. **We selected ResNet34 as a baseline to ensure a fair comparison with existing methods**, aligning with standard practices in the field.
>
> | Backbone | INATURALIST FPR95$\downarrow$   | INATURALIST AUROC$\uparrow$  | PLACES FPR95$\downarrow$   | PLACES AUROC$\uparrow$  | SUN FPR95$\downarrow$   | SUN AUROC$\uparrow$  | TEXTURES FPR95$\downarrow$   | TEXTURES AUROC$\uparrow$  | Average FPR95$\downarrow$   | Average AUROC$\uparrow$  |
> |----------|--------------------|-------------------|--------------|--------------|-----------|-----------|----------------|----------------|---------------|---------------|
> | Resnet34 | 18.62              | 95.89             | 26.70        | 94.49        | 20.06     | 95.52     | 57.90          | 85.36          | 30.82         | 92.81         |
> | Vit-b16  | 29.09              | 96.19             | 29.42        | 94.70        | 27.52     | 95.09     | 23.74          | 95.61          | 27.44         | 95.40         |
> | Convnext | 26.44              | 95.38             | 32.04        | 93.92        | 28.65     | 94.91     | 21.84          | 95.34          | 27.24         | 94.89         |
>
>
>
> # Q4 Applying the Unified OOD Score to Other Methods
>
> (Question 2)
>
> We appreciate the reviewer’s insightful concern regarding the contributions of outlier synthesis and OOD scoring in our framework.
>
> To address this, we contacted the authors of DreamOOD and obtained their synthesized OOD samples. Using their code, we retrained the detector and reproduced the results (FPR95: 39.39 vs. 38.76; AUROC: 91.81 vs. 92.02), confirming the consistency of our reimplementation. We then compared both the **default energy-based score** (used in DreamOOD, NCIS, BOOD, and other OE methods) and our **unified OOD score** under matched scoring conditions. The results are summarized below:
>
> | Best      | Unified OOD Score | INATURALIST FPR95$\downarrow$ | INATURALIST AUROC$\uparrow$  | PLACES FPR95$\downarrow$ | PLACES AUROC$\uparrow$  | SUN FPR95$\downarrow$ | SUN AUROC$\uparrow$  | TEXTURES FPR95$\downarrow$ | TEXTURES AUROC$\uparrow$  | Average FPR95$\downarrow$ | Average AUROC$\uparrow$  |
> | --------- | ----------------- | ----------------- | ----------------- | ------------ | ------------ | --------- | --------- | -------------- | -------------- | ------------- | ------------- |
> | Dream-OOD | ×                 | 23.86             | 96.12             | 38.49        | 93.26        | 39.86     | 93.07     | 55.34          | 84.81          | 39.39         | 91.81         |
> | GOOD      | ×                 | 18.62             | 95.89             | 26.70        | 94.49        | 20.06     | 95.52     | 57.90          | 85.36          | 30.82         | 92.81         |
> | Dream-OOD | √                 | 18.41             | 96.88             | 36.12        | 93.61        | 35.25     | 93.73     | 15.62          | 95.26          | 26.35         | 94.87         |
> | GOOD      | √                 | **9.34**          | **97.60**         | **24.82**    | **94.82**    | **17.30** | **96.10** | 19.32          | **96.60**      | **17.70**     | **96.28**     |
>
>
> These results show that:
>
> * **Without the unified score**, GOOD still outperforms DreamOOD with only **\~20%** of its synthesized samples, emphasizing the efficiency of our synthesis component.
> * The **unified score** provides a significant boost, especially on challenging datasets like Textures, where energy-based likelihood struggles due to overlap with the ID set. KNN distance improves separability, while on datasets like iNaturalist and SUN, energy-based likelihood is more discriminative.
>
> While the unified OOD score can also enhance exsiting methods, in GOOD it is **tightly coupled with the synthesis process**, both driven by the same underlying signals. This coupling enables us to **close the loop between generation and detection**, resulting in a more integrated and generalizable approach to OOD detection.
>
>
> # Q5 Lower Performance on PLACES365 Compared to FodFoM
>
> (Question 1)
>
> Thank you for your insightful question regarding the performance difference between GOOD and FodFoM on the PLACES365 dataset.
>
> **PLACES365** is a large-scale, diverse dataset with 365 scene categories, making it a challenging benchmark for OOD detection. **FodFoM** performs particularly well on this dataset due to its use of **BLIP-2 for generating detailed textual descriptions and GroundingDINO for creating background images**. These synthetic outliers closely match the background characteristics of PLACES365, leading to stronger performance. However, FodFoM’s approach is more limited on other datasets, as its focus on background generation and textual descriptions does not capture a wide range of OOD scenarios.
>
> In contrast, GOOD generates a broader range of OOD samples by leveraging both **image-level variations** and **feature-level sparsity**, leading to more generalized anomalies. This makes GOOD less aligned with PLACES365, resulting in comparatively lower performance on this dataset. However, GOOD offers a more **generalized approach** to OOD detection across various datasets, ensuring robustness in real-world applications.
>
> We will revise the manuscript to clarify these dataset-specific variations, highlighting FodFoM’s strengths on PLACES365 and GOOD’s broader applicability.
>
>
> # Q6 Clarity of the TFG Framework in the Main Text
>
> (Weakness 1)
>
> We appreciate the reviewer’s feedback. To enhance clarity, we will revise Section 3.2 to include a more detailed description of the TFG framework, covering its motivation, components, and workflow, in addition to the explanation in Appendix A.1. This will help readers better understand the framework's role without solely relying on the appendix.

---

> > ### Author Response · Authors · 2025-08-04
> > **Kind Follow-Up on Rebuttal**
> >
> > We are truly grateful for the time and thoughtful feedback you have provided. Our rebuttal addresses all the points raised, and we would greatly appreciate any further questions or suggestions. Your guidance is highly valued.

---

### Official Review · Reviewer_7K8B · 2025-07-02

**Clarity:** 2
**Significance:** 2
**Originality:** 2
**Rating:** 4
**Confidence:** 4

**Summary:**

This paper proposes a framework, GOOD, for synthesizing outliers for outlier-exposure based OOD detection. It generates OOD samples using text-to-image diffusion through both classifier-based pixel-level guidance and KNN-based feature-level guidance. To improve performance, GOOD uses varying sampling parameters when synthesizing samples, and uses a combination of likelihood- and distance-based OOD score at test time. Experimental results show that GOOD achieves competitive performance compared to existing methods.

**Questions:**

* For ablation studies in Table 3, it would be helpful to have more clarifications on what configuration each row corresponds to exactly. For (2) and (3), do they include the balanced sampling and unified OOD score components (i.e. everything without image/feature guidance), or are they ID classifier + outlier exposure with image/feature guidance samples? For row (4), is it the full GOOD framework (5) without the unified OOD score (the number seems to correspond to a=0 in Figure 6(b))?
* GOOD involves two distinct and somewhat orthogonal aspects, outlier synthesis and OOD scoring. This could make it hard to draw direct comparisons with various existing methods by just looking at the final performance: For instance, baselines like DreamOOD focus primarily on outlier synthesis, and default to simple scorings like using the OOD classifier output for OOD detection. Meanwhile, existing works like [1] and [2] have looked into combining multiple metrics for OOD scoring, although in a different setting. Since the paper seems to be motivated from the perspective of outlier synthesis methodology, it would be helpful to see direct comparisons with the baselines by holding the scoring function constant and using different synthesized outliers, to isolate the effect of synthesizing quality.

    * Specifically, what prompted this question is that in Figure 6(b), the major performance gain comes from the combination of energy and distance based OOD score. When a=0 (i.e. only free energy is used), GOOD’s performance seems only marginally better than the baselines. This raises some concerns of how much the proposed synthesis method helps vs. just combining multiple OOD metrics.

[1] Bergamin et al. "Model-agnostic out-of-distribution detection using combined statistical tests." AISTATS 2022.

[2] Park et al. "Nearest neighbor guidance for out-of-distribution detection." ICCV 2023.

**Ethical Concerns:**

["NO or VERY MINOR ethics concerns only"]

**Final Justification:**

Thank the authors for the rebuttal and the additional experiments. It addresses my concerns and therefore I'm raising my score to borderline accept.

**Limitations:**

Yes the authors have discussed limitations.

**Paper Formatting Concerns:**

No major formatting concerns

**Quality:**

2

**Strengths And Weaknesses:**

Strengths:
* The proposed method is technically sound.
* Experimental results show that the proposed method attains strong and consistent performance compared to baselines.
* Ablation studies are extensive, and analysis and visualizations provide intuitive ways of understanding the proposed method.

Weaknesses:
* The technical novelty is somewhat limited, where various individual components of the proposed framework are built upon existing techniques. For instance, the outlier synthesis part involves a relatively straightforward application of diffusion guided generation, and the unified OOD score is an ensemble of likelihood and KNN distance, two well-established OOD scoring methods.
* GOOD involves two orthogonal aspects, outlier synthesis and OOD scoring (the unified score). This could make comparison with some baselines like DreamOOD not immediately straightforward, as they focus primarily on outlier synthesis and default to simple metrics for OOD scoring. It would be helpful to see experiments and analysis with more direct and disentangled comparisons, e.g. by holding the OOD scoring part constant and isolating the effect of the outlier synthesis quality. (More details in Question 2 below.)

---

> ### Author Rebuttal · Authors · 2025-07-31
>
> # Q1 Clarification on the Ablation Study Configurations and Controls
>
> (Question 1)
>
> We apologize for any lack of clarity in the configuration details of Table 3. Below is a clearer breakdown of each row’s design:
>
> * **Row (2)** and **Row (3)** show **ID classifiers trained with outlier exposure**, where synthetic OOD samples are generated using **one type of guidance**—image-level (Row 2) or feature-level (Row 3). Both configurations use **balanced sampling across varying guidance strengths** to promote diversity in synthesized OOD data. However, these rows **do not include the unified OOD score** and rely solely on the default energy score for detection.
> * **Row (4)** represents the **full GOOD pipeline without the unified OOD score**, where both image-level and feature-level guidance are used during sampling, but the energy score alone is used for detection (matching **a = 0** in Figure 6(b)). This setting mirrors prior methods such as DreamOOD and BOOD.
> * **Row (5)** corresponds to the **full GOOD framework**, where both synthesis and scoring leverage the joint use of image- and feature-level signals for more effective OOD detection.
>
> The results highlight the complementary benefits of image- and feature-level guidance. Even without the unified OOD score (Rows 2-4), combining these signals outperforms prior state-of-the-art methods. The full GOOD framework (Row 5), where both synthesis and detection are guided by these signals, achieves the highest performance, providing a robust solution for OOD detection.
>
> We will revise the text and caption of Table 3 in the final version to clearly specify these configurations and avoid ambiguity. We appreciate the reviewer’s attention to detail, which helps improve the clarity of our presentation.
>
> # Q2 Disentangling the Impact of the Unified OOD Score
>
> (Question 2, Weakness 2)
>
> We appreciate the reviewer’s concern regarding the contributions of outlier synthesis and OOD scoring in our framework.
>
> To address this, we contacted the authors of DreamOOD and obtained their synthesized OOD samples. Using their code, we retrained the detector and reproduced the results (FPR95: 39.39 vs. 38.76; AUROC: 91.81 vs. 92.02), confirming our reimplementation. We then applied both the **default energy-based score** (used in DreamOOD, NCIS, BOOD, and many OE methods) and **our unified OOD score** for a direct comparison under matched scoring conditions. The results are summarized below:
>
> | Best      | Unified OOD Score | INATURALIST FPR95 | INATURALIST AUROC | PLACES FPR95 | PLACES AUROC | SUN FPR95 | SUN AUROC | TEXTURES FPR95 | TEXTURES AUROC | Average FPR95 | Average AUROC |
> | --------- | ----------------- | ----------------- | ----------------- | ------------ | ------------ | --------- | --------- | -------------- | -------------- | ------------- | ------------- |
> | Dream-OOD | ×                 | 23.86             | 96.12             | 38.49        | 93.26        | 39.86     | 93.07     | 55.34          | 84.81          | 39.39         | 91.81         |
> | GOOD      | ×                 | 18.62             | 95.89             | 26.70        | 94.49        | 20.06     | 95.52     | 57.90          | 85.36          | 30.82         | 92.81         |
> | Dream-OOD | √                 | 18.41             | 96.88             | 36.12        | 93.61        | 35.25     | 93.73     | 15.62          | 95.26          | 26.35         | 94.87         |
> | GOOD      | √                 | **9.34**          | **97.60**         | **24.82**    | **94.82**    | **17.30** | **96.10** | 19.32          | **96.60**      | **17.70**     | **96.28**     |
>
>
> Key takeaways:
> * **Without the unified score**, GOOD still outperforms DreamOOD with only **\~20%** of its synthesized samples, demonstrating the efficiency of our synthesis approach.
> * The **unified score** significantly boosts performance, especially on datasets like *Textures*, where the energy score struggles due to distribution overlap with the ID set. KNN in feature space improves separability in such cases, while the energy score excels in others, like *iNaturalist* and *SUN*.
>
> These results, supported by distribution analysis in Appendix A.3.3, motivate our unified score: different OOD shifts manifest in different spaces, and no single metric captures them all. We found: **Image-level likelihood** (via energy) captures pixel-space density mismatches. **Feature-level sparsity** (via KNN) is sensitive to semantic misalignment in embedding space. **Rather than treating these signals independently, we unify them—both in synthesis (via gradient guidance) and detection (via score fusion)—into a coherent, model-native framework**. This synergy is not just an add-on but **a reflection of our broader view of OOD detection**: that general OOD behavior should be modelled **holistically**, not piecemeal.
>
> We thank the reviewer for raising this important point and will revise the manuscript to more explicitly highlight this analysis.
>
> # Q3 Novelty of combining Image-Level and Feature-Level Signals
>
> (Question 2, Weakness 1)
>
> Our contribution extends well beyond the simple combination of image-level and feature-level signals for OOD detection. The **core innovation** in our work lies in using these two complementary signals—not just as detection cues, but as **proxies for defining general OOD behavior**, and most importantly, as **gradient-based guidance signals to steer generation**.
>
> This novel approach enables us to **bridge discriminative and generative paradigms without requiring additional training**. We leverage the gradients of both signals from a pretrained discriminative classifier to guide the sampling trajectory of a diffusion model, **expanding the generation boundary beyond the ID manifold**. The image-level signal informs how likely a sample is under the ID distribution, while the feature-level signal measures how far a sample is from semantic clusters in feature space. Their gradients pull the diffusion model toward **low-density, semantically off-manifold regions**, enabling the synthesis of realistic near-OOD samples without the need for latent space alignment or adversarial optimization.
>
> Our method does not treat these signals independently. Instead, it forms a **coherent, tightly coupled framework** in which:
> 1. Both signals jointly guide the generation process,
> 2. Balanced sampling over varying anomaly strengths ensures diversity in the synthesized OOD samples, and
> 3. The same signals are reused during detection through a unified scoring mechanism.
>
> This integrated design ensures that the guidance used during generation directly aligns with the scoring function used during detection, resulting in a **theoretically grounded and practically robust OOD detection pipeline**.
>
> In summary, our work presents a **principled and generalizable framework** that tightly couples discriminative modeling, generative sampling, and dual-signal detection. By leveraging the **guided generative capability of discriminative classifiers**, we enable training-free synthesis of diverse OOD samples—without needing retraining, latent projections, or external datasets.
>
> We thank the reviewer for emphasizing the **long-term impact and generalizability** of OOD detection research. We believe our framework offers a **well-motivated and unified approach**, bridging generative and discriminative paradigms by utilizing model-native signals to define and expose OOD structure. This perspective has the potential to inspire future research into safe, robust, and generalizable open-world learning.

---

> > ### Author Response · Authors · 2025-08-04
> > **Kind Follow-Up on Rebuttal**
> >
> > We are truly grateful for the time and thoughtful feedback you have provided. Our rebuttal addresses all the points raised, and we would greatly appreciate any further questions or suggestions. Your guidance is highly valued.

---

> > > ### Comment · Reviewer_7K8B · 2025-08-05
> > >
> > > Thank the authors for the rebuttal and the additional experiments. They address my concerns and I'm content with the response. Therefore I'm raising my score to borderline accept.

---

### Official Review · Reviewer_e319 · 2025-07-02

**Clarity:** 3
**Significance:** 3
**Originality:** 3
**Rating:** 5
**Confidence:** 4

**Summary:**

This paper proposes an OOD-detection method for classifiers based on two fundamental signals - the *free energy* or log-sum-exp of a classifier's logits, and proximity to in-distribution data in the model's intermediate embedding spaces. These signals are operationalized in 2 high-level ways:
1. As guidance terms for the generation of outliers from a pretrained diffusion model and
2. As individual scores which can be combined into an OOD detection metric.

To build an OOD detector, the authors first train a model using outlier exposure on samples generated by (1), and then detect samples using the OOD score (2). This combo yields near-uniform outperformance across ID-OOD dataset pairs.

**Questions:**

You have demonstrated convincingly that both guidance strategies as well as the unified OOD score are necessary for strong performance. However, I don't see much in the way of high-level takeaways from these components. How can your insights here guide the design of future OOD detection algorithms?

**Ethical Concerns:**

["NO or VERY MINOR ethics concerns only"]

**Final Justification:**

All of my concerns were addressed throughout the rebuttal period. It's a good paper containing novel ideas and strong performance across baselines.

**Limitations:**

Yes

**Quality:**

3

**Strengths And Weaknesses:**

Strengths
- A lot of care is taken to explain the idea through detailed imagery and clear writing.
- Well-designed experiments and ablation studies.
- The method is carefully designed and performs well across the board. In my opinion, there is genuine novelty here and it's tough to argue with these results.

Weaknesses
- Your given likelihood-based motivation needs to be recalibrated a bit for GOOD_img  and the $E$ score. Your comment (L114) that "the negative log-sum-exp of the logits ... is proportional to the negative log-likelihood of the input" is not true - one needs to train a discriminative model in a very specific way for this proportionality to hold (e.g. as in [A]). (The NeurIPS 2020 edition of your citation [34] makes this claim erroneously, and its April 2021 arXiv update has been revised to remove this claim.) Your argument that this term somehow represents likelihood is thus dubious.
- This paper is something of a bag of tricks and isn't firmly grounded on generalizable insights for OOD detection. Performance here is very good but the ideas in this work might have a short half-life. More extensive exposition in the intro, experiments, and conclusion about the big picture ideas driving your work, e.g. about why both "image-level" and "feature-level" signal is necessary for good OOD detection, should help here.

Once these weaknesses are adequately addressed, I'll raise my score.

[A] Grathwohl, Will, et al. "Your classifier is secretly an energy based model and you should treat it like one." _International Conference on Learning Representations_.

---

> ### Author Rebuttal · Authors · 2025-07-31
>
> # Q1 Validity of the correlation between Free Energy and Likelihood
>
> (Weakness 1)
>
> We would like to clarify the potential misunderstanding. Our use of the **free energy** term $E(x ; f)=-\log \sum_i e^{f_i(x)}$ is not intended as a direct representation of likelihood, but rather as a scoring function to differentiate between ID and OOD samples. It has a well-established relationship with the log-likelihood $\log p(x)$, which is supported by a solid theoretical foundation, as shown in [A], [B] (LeCun et al., 2006), and the ICLR paper referenced by the reviewer [C].
>
> In [B], the foundation of energy learning is established, demonstrating that any probability density $p(\mathbf{x})$ for $\mathbf{x} \in \mathbb{R}^D$ can be expressed as:(also Eq. (2) in [C]) $$
> p_\theta(\mathbf{x})=\frac{\exp \left(-E_\theta(\mathbf{x})\right)}{Z(\theta)}
> $$where $E_\theta(\mathbf{x})$ is the energy function (not to be confused with free energy). The JEM framework [C] reinterprets a classifier as a joint energy-based model over $(x, y)$. Begin with a classifier (Eq(4) in their paper) : $$ p_\theta(y \mid \mathbf{x})=\frac{\exp \left(f_\theta(\mathbf{x})[y]\right)}{\sum_{y^{\prime}} \exp \left(f_\theta(\mathbf{x})\left[y^{\prime}\right]\right)} $$where $f_\theta(\mathbf{x})[y]$ denotes the $y^{\text {th }}$ index of $f_\theta(\mathbf{x})$, i.e., the logit corresponding to the $y^{\text {th }}$ class label. A key observation in their work is that the logits from $f_\theta$ can be reinterpreted to define $p(\mathbf{x}, y)$ and $p(\mathbf{x})$ as well. Without altering $f_\theta$, one can reuse these logits to define an energy-based model of the joint distribution of $\mathbf{x}$ and $y$ as in (Eq(5) in their paper, following Eq. (2)):$$ p_\theta(\mathbf{x}, y)=\frac{\exp \left(f_\theta(\mathbf{x})[y]\right)}{Z(\theta)} $$where $Z(\theta)$ is the normalizing constant and $E_\theta(\mathbf{x}, y)=-f_\theta(\mathbf{x})[y]$.
>
> By marginalizing out $y$, we arrive at an unnormalized density model for $\mathbf{x}$ as described in (Eq(6) in their paper):$$ p_\theta(\mathbf{x})=\sum_y p_\theta(\mathbf{x}, y)=\frac{\sum_y \exp \left(f_\theta(\mathbf{x})[y]\right)}{Z(\theta)} . $$This allows us to conclude that $p_\theta(\mathbf{x}) \propto \sum_y \exp \left(f_\theta(\mathbf{x})[y]\right)$, establishing a clear connection between energy functions and probability densities. While the link between energy and likelihood may seem indirect, in practice, the energy score is still **proportional to the negative log-likelihood**, a relationship that is well-supported by existing literature. Additionally, [C] aims to optimize the joint distribution $\log p(x,y)=\log p(x)+\log (y|x)$ to obtain both a generative model and a discriminative model. Their statement, "... optimize $\log p(x)$ using Equation (2) with SGLD, where gradients are taken with respect to $LogSumExp_{y}(f_{\theta}(x)[y])$...", further reinforces the strong connection between likelihood and energy, aligning with the gradient-based intuition presented in our paper.
>
> In contrast, in [A] and our paper, we directly use the (negative) log-sum-exp of the logits—$E(x;f) = -\log \sum_i e^{f_i(x)}$. This term can be derived from a purely discriminative classifier for OOD detection, without the need to learn a normalized density model. This approach avoids the challenging partition-function normalization that is typical in generative energy-based models. The energy score for OOD detection is theoretically aligned with the input's probability density—samples with higher energies correspond to data points with a lower likelihood of occurrence [A, page 2, lines 1-2]. This can be rigorously proven by [B] (Page 14-15), where the theoretical basis for bridging classifiers and energy models is laid out. Specifically, the discriminative model trained with negative log-likelihood (NLL) loss will reduce the energy for in-distribution data points.
>
> We appreciate your attention to these details and your careful review of the relevant literature. To ensure full clarity, we will include a more detailed explanation and formal derivation of these principles in the appendix.
>
> **References:**
>
> [A] Liu, Weitang, et al. "Energy-based out-of-distribution detection." *Advances in Neural Information Processing Systems*, 33 (2020): 21464-21475.
>
> [B] LeCun, Yann, et al. "A tutorial on energy-based learning." *Predicting Structured Data*, 1.0 (2006).
>
> [C] Grathwohl, Will, et al. "Your classifier is secretly an energy based model and you should treat it like one." *International Conference on Learning Representations*, 2020.
>
>
> # Q2 Generalizable Insights for Future OOD Detection Algorithms
>
> (Weakness 2)
>
> Our contribution goes beyond simply combining image-level and feature-level signals for OOD detection. The **core innovation** of our work is using these two complementary signals—not only as detection cues but as **proxies for defining general OOD behavior** and, importantly, as **gradient-based guidance signals for steering generation**.
>
> This approach enables us to **bridge discriminative and generative paradigms in a training-free manner**. Specifically, we leverage the gradients of both signals from a pretrained discriminative classifier to guide the sampling trajectory of a diffusion model, **expanding the generation boundary beyond the ID manifold**. The image-level signal captures how likely a sample is under the ID distribution, while the feature-level signal indicates how far a sample is from semantic clusters in the feature space. These gradients pull the diffusion model toward **low-density, semantically off-manifold regions**, allowing us to generate **challenging and realistic near-OOD samples** without requiring latent space alignment or adversarial optimization.
>
> Our method treats these signals **as a tightly coupled framework** where:
> 1. Both signals jointly guide the generation process,
> 2. Balanced sampling ensures diverse OOD samples across varying anomaly strengths,
> 3. The same signals are reused during detection through a unified scoring mechanism.
>
> This unified design ensures that the guidance used during generation aligns directly with the scoring function used during detection, resulting in a **theoretically grounded and robust OOD detection pipeline**.
>
> In summary, we introduce a **principled, generalizable framework** that integrates discriminative modeling, generative sampling, and dual-signal detection. By unlocking the **generative capability of discriminative classifiers**, our method enables **training-free synthesis of diverse OOD samples**, eliminating the need for retraining, latent projections, or external datasets.
>
> We sincerely appreciate the reviewer’s focus on the **long-term impact and generalizability** of OOD detection research. We believe our framework offers a **well-motivated, unified approach** that bridges generative and discriminative paradigms, defining and exposing OOD structure with model-native signals. We hope this perspective will inspire future research into safe, robust, and generalizable open-world learning.

---

> > ### Comment · Reviewer_e319 · 2025-08-02
> >
> > Further on Q1:
> >
> > My understanding is that you are claiming that for an arbitrary pretrained classifier $f_\theta$, the sum of exponentiated logits $\sum_y e^{f_\theta(x)[y]}$ represents some approximation of an energy function $E(x)$ such that $p(x) = e^{-E(x)}/C$, where $p(x)$ is the ground truth density. This is not true. If this is not what you are claiming, can you please clarify the connection you’re making between $\sum_y e^{f_\theta(x)[y]}$ and $p(x)$?
> >
> > The sources you gave do not support the claim at all:
> > - Lecun et al. (2006) pg 14-15 proves no such thing.
> > - Grathwohl et al. (2020) and the description you wrote out show that you can reinterpret $\sum_y e^{f_\theta(x)[y]}$ as a model for $E(x)$ *only by training your network specifically to satisfy this property* (i.e., by training it as a joint energy model). The entire contribution of their paper derives from the fact that this is not true for most classifiers - $\sum_y e^{f_\theta(x)[y]}$ is usually a meaningless quantity.
> > - The claim that this property holds for non-JEMs has been scrubbed from the most recent arxiv version of Liu et al. (2020), probably because the authors have realized it is incorrect.

---

> ### Author Response · Authors · 2025-08-03
> **On the Validity of Energy-Based Interpretation in Our Method**
>
> Thank you very much for your patient, detailed, and professional question regarding the validity and interpretation of the energy-based formulation used in our work. We are happy to provide further clarification on these points, and we hope our explanation will help address your concerns and provide more insight into our design choices.
>
> **1. What we claim (and what we do not).**
>
> Given any classifier with logits $f_\theta(x)[y]$, define the energy:
> $$
> E_\theta(x,y) := -f_\theta(x)[y], \quad
> E_\theta(x) := -\log\sum_y e^{f_\theta(x)[y]}.
> $$ These definitions induce the (model) joint and marginal densities:
> $$
> p_\theta(x,y) = \frac{e^{f_\theta(x)[y]}}{Z(\theta)}, \quad
> p_\theta(x) = \sum_y p_\theta(x,y) = \frac{\sum_y e^{f_\theta(x)[y]}}{Z(\theta)} = \frac{e^{-F_\theta(x)}}{Z(\theta)},
> $$ where $Z(\theta)$ is the parameter-dependent partition function (constant in $x$). **These are identities that hold for any $\theta$ once one chooses to interpret the logits as an energy**.
>
> **2. Clarification on the Term "energy"**
>
> Following LeCun et al. (2006), any scalar-valued function over the input variable can define an energy-based model (P1, also P2 in Grathwohl et al. (2020)). Thus, the logit $f_\theta(\mathbf{x})[y]$ can be interpreted as an energy $E(\mathbf{x}, y) = -f_\theta(\mathbf{x})[y]$. The aggregated quantity $\log \sum_y \exp(f_\theta(\mathbf{x})[y])$, commonly referred to as **free energy**, is another valid energy function over $\mathbf{x}$. In our work (and in Liu et al., 2020), we adopt the free energy for scoring and guidance. The former one is still useful in intermediate derivations, as shown in Grathwohl et al. (2020) (Equation (4)).
>
> **3. On Grathwohl et al. (2020, JEM).**
>
> Grathwohl et al. (2020) demonstrate that the logits of a standard classifier can be **reinterpreted** to define a joint density over inputs and labels via:
> $$
> p_\theta(x, y) = \frac{e^{f_\theta(x)[y]}}{Z(\theta)}, \quad
> p_\theta(x) = \sum_y p_\theta(x, y) = \frac{\sum_y e^{f_\theta(x)[y]}}{Z(\theta)}.
> $$ **These identities are purely algebraic results of the energy-based parameterization and hold regardless of how the model is trained. JEM leverages this reinterpretation to train the network as a joint energy model**, thereby unifying generative and discriminative modeling**. However, the validity of the above equations does not depend on performing such training. The training procedure in JEM optimizes the model under this interpretation; it does not retroactively make these identities true—they follow directly from the form of the logits.
>
> **4. On Liu et al. (2020).**
>
> Liu et al. (2020) provide a complementary perspective by showing that a model trained with negative log-likelihood (NLL) loss will reduce the free energy for in-distribution data points. This result is derived via gradient-based analysis, a technique also adopted by LeCun et al. (2006) in the context of energy-based learning.
>
> Notably, across all versions of Liu et al.'s arXiv submission (v1 to v4), this core conclusion remains unchanged. As emphasized in their abstract, it forms the theoretical foundation of their work: “energy scores are theoretically aligned with the probability density of the inputs,” and “energy can be flexibly used as a scoring function for any pre-trained neural classifier.”
>
> **5. Implications for our method.**
>
> Our approach only requires that **$E_\theta(x)$ be a well-defined, monotone transform of an unnormalized log-density (up to the constant $Z(\theta)$)**, so that its value serves as a score and its gradient $\nabla_x E_\theta(x)$ provides a meaningful guidance signal. We never use or estimate $Z(\theta)$, and we never claim that $p_\theta(x)$ equals the ground-truth $p(x)$ for arbitrary pretrained classifiers.
>
> **6. Text changes we will make.**
>
> To avoid any ambiguity, we will revise the paper to:
> -  replace phrases that might suggest likelihood with “**unnormalized log-density up to a parameter-dependent constant**” and “**energy/free-energy score**”;
> -  explicitly state that JEM-style training is required only if one seeks a calibrated generative model, which our method does not;
> -  cite LeCun et al. for the EBM formalism, Grathwohl et al. for the reinterpretation and training objective, and Liu et al. for using energy/free-energy as a classifier-agnostic OOD score.
>
> We hope this clarifies that our use of $\sum_y e^{f_\theta(x)[y]}$ follows a standard construction in energy-based modeling, and provides a sound and practical basis for both scoring and gradient-based guidance in our method. We would like to express our sincere thanks once again, and warmly welcome any further questions or discussions. We truly appreciate and value such a professional and in-depth exchange.

---

> ### Comment · Reviewer_e319 · 2025-08-04
>
> Hello,
>
> Thanks for describing your position in full. The distinction between $p_\theta$ and $p$ is a helpful point of clarification, and I'm glad we're on the same page. For the rest of this comment, I'll replace $p$ with $p_\theta$ in quotes where appropriate for clarity, and use $p$ to refer to the ground truth density.
>
> Thanks for the suggested changes, but I'm going to suggest some more specific ones. The claims I take issue with mostly appear in Section 3.1. For example,
> > We begin with a crucial observation: the denominator of the softmax predictive distribution is closely related to the input’s marginal likelihood. For a classifier $f_\theta$, the softmax probability for class $y$ given input $x$ is expressed as $p_\theta(y \mid x) = $... **This formulation indicates** that the negative log-sum-exp of the logits, also known as the free energy, is proportional to **the** negative log-likelihood of the input ...
>
> It runs counter to your explanation above and is misleading for a few reasons:
> 1. "This formulation indicates" would suggest the definition of $p_\theta(x)$ (using the notation from your latest comment) somehow derives from the traditional softmax formulation of $p_\theta(y \mid x)$. It doesn't - it's a new density which you are defining in the subsequent line. This definition is the  "reinterpretation" proposed by Grathwohl et al. (2020).
> 2. In light of this fact, there's no "crucial observation" being made here. Of course the denominator is related to the "marginal likelihood" - you are defining the likelihood using denominator!
> 3. The phrases "**the** input’s marginal likelihood" "**the** negative log-likelihood of the input" when introduced out of nowhere using the article *the* indicate some quantity that already exists and is known, namely the ground truth log-likelihood $\log p(x)$ (more formally the log-density, but the term log-likelihood for this function is ubiquitous in ML). You should make clear that this is a definition of a new density, not some already-extant quantity.
> 4. Your Eq. (3) $E_\theta(x; f) := - \log \sum_{k=1}^C e^{f_k(x, \theta)} \propto - \log p_\theta(x)$ suggests that the definition being made is for $E_\theta(x; f)$ and that $- \log \sum_{k=1}^C e^{f_k(x, \theta)} \propto - \log p_\theta(x)$ is a relation known a priori. In fact, both the equality and the $\propto$ are definitions here!
>
> You then go on to comment that "ID samples tend to lie in high-likelihood regions", where "likelihood" is presumably $p_\theta(x)$. However, $p_\theta(x)$ is in no way equal to the ground truth density $p(x)$, so this is not at all an obvious supposition! The only thing that justifies this claim is the gradient-based argument of Liu et al. (2020), and you need to make this connection explicit.
>
> In general, throughout the paper, you need to
> 1. make clear which relation is a definition you are making and which one follows logically,
> 2. clearly differentiate the score you call $p_\theta(x)$ in your most recent response ($p(x)$ in your manuscript) from the ground truth density usually denoted $p(x)$ in particular  (your first suggested change around changing wording for $p_\theta(x)$ helps, but the distinction between $p_\theta(x)$ and $p(x)$ should be made explicit) , and
> 3. make clear why $p_\theta(x)$ is a useful OOD score (i.e., cite and clearly describe the conclusions of Liu et al. (2020)).
>
> Hopefully this helps.

---

> > ### Author Response · Authors · 2025-08-05
> > **Appreciation for Your Rigorous and Thoughtful Review**
> >
> > We would like to sincerely thank the reviewer once again for the careful analysis and the patient, constructive discussion. We believe this kind of engagement greatly contributes to a healthy and rigorous review culture in the community. We are also pleased that, through several rounds of clarification, we were able to identify the source of potential misunderstandings and ultimately reach a shared understanding.
> >
> > It is indeed true that $p_\theta(x)$ is not equal to the true density $p(x)$. The connection arises from the use of the negative log-likelihood (NLL) loss to train discriminative models, which corresponds to maximum likelihood estimation (MLE). We fully agree with your distinction between **definition** and **deduction**—this relationship is not a direct observation, but rather a consequence of a formal reinterpretation. Specifically, following Grathwohl et al. (2020), one must first redefine $p_\theta(x, y)$ as an energy-based model before deducing the proportionality between $p_\theta(x)$ and the energy score $E_\theta(x; f)$. Furthermore, as shown by Liu et al. (2020), gradient-based analysis is required to justify that in-distribution (ID) samples tend to lie in high-likelihood regions and exhibit lower free energy.
> >
> > Below we detail the changes we will make to resolve each of your concerns.
> > ## 1. “_This formulation indicates …_” and the claim of a “crucial observation”
> >
> > **Reviewer’s point.**
> > The current wording could give the impression that the softmax denominator _implies_ a pre-existing density $p_\theta(x)$; in fact we are **defining** that density à la Grathwohl et al. (2020).
> >
> > **Our action.**
> > - We will replace the sentence with:
> >     > “We **define** the energy (LeCun et al., 2006) with logits $f_y(x; \theta)$ of given classifier$$
> > E_\theta(x,y) := -f_y(x; \theta), \quad
> > E_\theta(x) := -\log\sum_y e^{f_y(x; \theta)}.$$ These definitions induce the (model) joint and marginal densities: $$
> > p_\theta(x,y) = \frac{e^{f_y(x; \theta)}}{Z(\theta)}, \quad
> > p_\theta(x) = \sum_y p_\theta(x,y) = \frac{\sum_y e^{f_y(x; \theta)}}{Z(\theta)} = \frac{e^{-E_\theta(x)}}{Z(\theta)},
> > $$ following the reinterpretation of Grathwohl et al. (2020).”
> > - Throughout the manuscript we will treat this relationship **explicitly as a definition** and discuss only the **logical consequences** that follow, rather than presenting it as a “crucial observation.”
> > ## 2. Use of the definite article “the” for new quantities
> >
> > **Reviewer’s point.**
> > Wording such as “**the** input’s marginal likelihood” implies that the quantity already exists.
> >
> > **Our action.**
> > - Throughout the manuscript we will replace “the input’s marginal likelihood” and “the negative log-likelihood of the input” with “_the **model-defined** marginal likelihood_ $p_\theta(x)$” and “_its negative log-density_ $-\log p_\theta(x)$”.
> > - A sentence will be added at first mention to state explicitly that these are **new model-dependent quantities**, distinct from the unknown ground-truth density $p(x)$.
> >
> > ## 3. Equation (3) and proportionality to $-\log p_\theta(x)$
> >
> > **Reviewer’s point.**
> > Presenting $$E_\theta(x;f) := -\log\sum_{k=1}^{C} e^{f_k(x,\theta)} \;\propto\; -\log p_\theta(x)$$ could be misread as a derived fact rather than a definition.
> >
> > **Our action.**
> > - We will rewrite Eq. (3) as two separate **definitions**:
> >     >$$E_\theta(x;f) := -\log\sum_{k=1}^{C} e^{f_k(x,\theta)}, \qquad p_\theta(x) = \frac{e^{-E_\theta(x;f)}}{Z(\theta)}.$$
> >     A short remark will clarify that the proportionality constant is absorbed in the normalisation of $p_\theta$.
> > ## 4. Justifying the claim that ID samples lie in high-$p_\theta$ regions
> >
> > **Reviewer’s point.**
> > The manuscript asserts that ID samples tend to fall in high-$p_\theta(x)$ regions, but this is not self-evident. The justification appears to rely on Liu et al. (2020), and the practical relevance of $p_\theta(x)$ as an OOD scoring function should be made more explicit.
> >
> > **Our action.**
> > - We will revise the sentence to:
> >     > “Following Liu et al. (2020), we note that ID samples tend to concentrate in regions of high $p_\theta(x)$ and low energy, as supported by their gradient-based analysis.”
> > - In Appendix B, we will provide a more detailed summary of this gradient-based analysis. It not only supports the use of energy as a score for distinguishing ID from OOD samples, but also offers theoretical grounding for the gradient-guided approach proposed in our work.
> >
> > We believe that through our joint effort, the theoretical foundations of the paper have become more solid and the exposition clearer—making it easier for readers to understand the key ideas without confusion. This, in our view, reflects the very purpose of the rebuttal process. We are truly grateful for the opportunity to improve the manuscript through this exchange, and we sincerely hope that our revisions address your concerns fully.

---

> > > ### Comment · Reviewer_e319 · 2025-08-06
> > >
> > > Thanks for engaging at length on this topic. This resolves all my concerns, so I'll raise my score to an accept.

---

### Official Review · Reviewer_Gz6x · 2025-07-03

**Clarity:** 3
**Significance:** 2
**Originality:** 3
**Rating:** 4
**Confidence:** 3

**Summary:**

This paper proposes a framework (GOOD) for OOD detection that synthesizes diverse and informative OOD samples by directly guiding the sampling trajectory of diffusion models. GOOD leverages classifiers to provide dual-level guidance: (1) image-level guidance based on input likelihood gradients, and (2) feature-level guidance using k-nearest neighbor distances in latent space. A unified OOD score combining both guidance signals is introduced for robust detection. Experiments on ImageNet-100 and CIFAR-100 demonstrate that training classifiers with GOOD-generated samples significantly improves OOD detection performance, achieving state-of-the-art results in FPR95 and AUROC across multiple datasets.

**Questions:**

1. It is unclear under which assumption that  the generated OOD sample can be regarded as real OOD samples and is there  any  evidence that the generated samples by the proposed methods are approximating OOD samples in some sense?  Also visually the generated ones are semantically close to Id data, so is there any analysis/insight that what kind of OOD samples can be effectively detected by this method?

2.	Can the authors provide a comparison of the efficiency of GOOD with other OOD detection methods? A direct evaluation of runtime or resource usage would help clarify the practical cost of the proposed approach.

3.   Can the method be validated on a real-world OOD detection task， for example a medical  image anomaly (rare-disease) application?

4.	Is it possible to introduce a general strategy for selecting the multiple hyperparameters involved in GOOD? Given the number of tunable hyperparameters, a general guideline would enhance the usability and robustness of the GOOD framework.

**Ethical Concerns:**

["NO or VERY MINOR ethics concerns only"]

**Final Justification:**

Thanks the authors for the efforts on the details response and adding test on real ODD scenario. Although I am not totally convinced by the novelty and the practical value of the proposed method, I raise my score to broadline accept.

**Limitations:**

My main concerns are outlined in the questions above.

**Quality:**

2

**Strengths And Weaknesses:**

Strengths:
1.The paper is well-written and clearly structured, with comprehensive figures.
2.The method introduces a combination of image-level and feature-level guidance for diffusion sampling, for generating OOD samples.
3.Experiments on benchmark datasets show that GOOD achieves state-of-the-art results.

Weakness:

1.  The method is validated on the toy examples for OOD detection. Can it be validated on any real OOD application scenario?

2. The performance of the synthesized OOD samples depend heavily on the capabilities of the pretrained diffusion model and the classification method. In particular, these models may struggle to generate OOD samples for certain domains. This likely explains its relatively lower performance on the Textures dataset.

3. The method involves several tunable hyperparameters—such as guidance strengths, condition strength, k-NN neighborhood size, and OOD loss weight. While the authors conduct ablation studies to analyze their effects, the overall sensitivity to these hyperparameters remains relatively high and may hinder generalization across tasks without careful tuning.

4. Despite being training-free, GOOD requires diffusion sampling, which is computationally expensive. This may limit its practicality in low-resource settings. It would be helpful if the authors provided a detailed comparison of inference time or computational cost with other OOD detection methods.

---

> ### Author Rebuttal · Authors · 2025-07-31
>
> # Q1 Assumption and insights of Generated OOD Samples
>
> (Question 1)
>
> We define OOD based on the inherent properties of the model and data:
> 1. **Image-level likelihood**—OOD lies in low-density regions under the ID model;
> 2. **Feature-level separation**—OOD is distant from ID semantic clusters in the feature space.
>
> These criteria align with two well-established families of OOD scores—**logits-based** (confidence, energy) and **feature-based** (KNN distance) \[A, B]. Prior work has also shown that OOD can result from **semantic**, **covariate**, and **domain shifts** \[C]. Building on these insights, we guide the generator using two differentiable OOD signals, directly reflecting these criteria.
>
> Regarding the statement "visually the generated samples are semantically close to ID data," this is rational, as we aim to generate challenging **near-OOD** samples. Exhaustively synthesizing all real OOD modes is impractical, so instead, we move just beyond the ID boundary to generate near-OOD samples—may be visually similar to ID data but in **low-likelihood, off-manifold** regions. Training the classifier to distinguish ID from these hard samples sharpens the decision boundary, making it more sensitive to unseen real OOD samples.
>
> On ImageNet (Figures 4 and 8), our distributional analyses confirm the design's effectiveness. Both generated OOD and real OOD samples show clear separation from ID based on likelihood and feature-distance scores, aligning with our goal of sampling just outside the ID boundary. The exception is **Textures**, where the energy score overlaps more with ID, reducing its discriminative power. However, adding feature-level distance restores separation, showing that the combined guidance is robust across OOD types.
>
>
> Ref:
> [A] Liu et al. Energy-based out-of-distribution detection. Neurips 2020
> [B] Sun et al. Out-of-distribution detection with deep nearest neighbors. ICML 2022
> [C] Yang et al. Generalized out-of-distribution detection: A survey. IJCV 2024
>
> # Q2 Practical Efficiency and Computational Cost
>
> (Question 2, Weakness 4)
>
> We provide a detailed **computational cost comparison** in Appendix A.3.2. Existing outlier exposure methods can generally be categorized into two approaches: (1) **data collection-based** and (2) **generation-based**. The former, such as WOODS and SAL, rely on manually curated external datasets, which are costly to collect, often domain-specific, and difficult to scale. The latter, including Dream-OOD, FodFoM, NCIS, and BOOD, use **multi-stage generation pipelines** that involve significant computational and engineering overhead.
>
> | Method          | Manual Data Collection | ID Embedding Alignment | OOD Embedding Sampling | OOD Sample Generation | Training and Detection |
> | --------------- | ---------------------- | ---------------------- | ---------------------- | --------------------- | ---------------------- |
> | WOODS     | ✓                      |                        |                        |                       | ✓                      |
> | SAL        | ✓                      | ✓                      |                        |                       | ✓                      |
> | Dream-OOD |                        | ✓ (8.2 h)              | ✓                      | ✓ (10.1 h)            | ✓ (8.5 h)              |
> | FodFoM     |                        | ✓                      | ✓                      | ✓                     | ✓                      |
> | NCIS       |                        | ✓ (13 h)               |                        | ✓                     | ✓                      |
> | BOOD      |                        | ✓ (0.62 h)             | ✓ (0.1 h)              | ✓ (7.5 h)             | ✓ (8.5 h)              |
> | **GOOD (Ours)** |                        |                        |                        | ✓ (\~7.8 h)                | ✓ (\~5.9 h)            |
>
> In contrast, GOOD simplifies the process by directly guiding diffusion sampling with gradients from a pretrained classifier, skipping latent space construction, embedding alignment, and OOD feature synthesis. It involves just two steps: **sampling and detection**, making the pipeline more efficient.
>
> Even without a pretrained classifier, GOOD remains efficient. For example, we trained a ResNet-34 on ImageNet-100 **from scratch** in **1.7 hours** and used it to guide OOD sampling. The OOD detector was **fine-tuned in 4.2 hours** on the same backbone following a continual learning paradigm.
>
>
> # Q3 Real-World Applicability on Medical images
>
> (Question 3, Weakness 1)
>
> Following the reviewer’s suggestion, we evaluate GOOD on a real clinical-style task. We use a **brain MRI tumor dataset** with three classes—**Pituitary**, **Meningioma**, and **Glioma**—and treat **Pituitary + Meningioma** as **ID** (n=1,884). We regard **Glioma** as **OOD‑1** (n=708) and add two additional real OOD sets: **Alzheimer’s brain MRI** (**OOD‑2**, n=1,280) and **intracranial bleeding CT** (**OOD‑3**, n=200). All datasets are publicly available on Kaggle; further details and dataset links will be provided in the Appendix.
>
> We follow our pipeline and use the publicly available medical diffusion model “Nihirc/Prompt2MedImage.” Using our two guidance signals, we generate **1,000 OOD images per guidance** (total **2,000** images).
>
> |                      | Glioma FPR95$\downarrow$ | Glioma AUROC$\uparrow$ | Alzheimer FPR95$\downarrow$ | Alzheimer AUROC$\uparrow$ | Bleeding FPR95$\downarrow$ | Bleeding AUROC$\uparrow$ | Average FPR95$\downarrow$ | Average AUROC$\uparrow$ |
> | -------------------------- | -------------------------------- | ------------------------------ | ----------------------------- | --------------------------- | ---------------------------- | -------------------------- | --------------------------- | ------------------------- |
> | w/o OOD images             | 82.06                            | 63.84                          | 14.06                         | 95.69                       | 77.00                        | 68.75                      | 57.71                       | 76.09                     |
> | + unified score        | 73.31                            | 84.95                          | 0.00                          | 97.70                       | 56.00                        | 87.05                      | 43.10                       | 89.90                     |
> | w/ our OOD images          | 72.18                            | 77.03                          | 0.00                          | 99.02                       | 39.00                        | 84.02                      | 37.06                       | 86.69                     |
> | + unified score (Ours) | **15.96**                        | **96.41**                      | **0.00**                      | **99.28**                   | **25.00**                    | **94.25**                  | **13.65**                   | **96.65**                 |
>
> **Results (Table above):**
> * Using generated OOD images alone improves robustness: avg. FPR95 drops from 57.71 to 37.06, and AUROC increases from 76.09 to 86.69 across three real-OOD datasets.
> * The unified score alone improves performance (baseline → “+ unified score”), but the greatest improvements occur when combined with GOOD-generated OOD images, especially for the challenging near-OOD case, Glioma.
>
> Qualitatively, as the OOD guidance strength increases, the generated images show **reduced lesion-like structures from ID classes**, **contrast/intensity redistribution** (resembling **modality or acquisition shifts**), **varying slice orientations**, and at high guidance, occasional off-manifold **artifacts**—all characteristic of OOD. These variations help the classifier learn a **sharper, more transferable decision boundary**, which explains its strong generalization to unseen real OOD in medical imaging.
>
>
> # Q4: Hyperparameter Sensitivity and Generalization
>
> (Question 4, Weakness 3)
>
> **GOOD does not require extensive hyperparameter tuning to perform well.** The key hyperparameters are the guidance strengths $\rho$ and $\mu$, the KNN distance parameter $k$, the regularization weight $\lambda$, and the coefficient $a$ for the unified OOD score.
>
> The most challenging parameters to tune are $\rho$ and $\mu$, as they control how much the generated samples deviate from the ID manifold. **Instead of searching for a single optimal setting, we use a balanced sampling strategy across values (e.g., {0.1, 0.5, 1, 2, 5}), as detailed in Section 3.2**. This avoids tedious tuning and helps construct a diverse OOD sample set, improving robustness and coverage.
>
> The coefficient $a$, which balances energy-based and kNN-based scores, impacts performance depending on OOD dataset separability. While tuning $a$ can boost performance, our method performs well without it, and the unified score is a refinement, not a core dependency.
>
> For other hyperparameters like $k$ and $\lambda$, a sensitivity analysis in Section 4.3 and Appendix A.2.2 shows that the method is robust to a wide range of values, with default settings working well across all datasets.
>
>
> # Q5: Challenges with Pretrained Diffusion Models
>
> (Weakness 2)
>
> Thank you for your insightful comment. The performance of our approach is indeed influenced by the pretrained diffusion model's capabilities. However, **our framework maximizes the use of existing generative models**, reducing the need for labor-intensive, biased OOD data collection, especially in the specific field. While performance on the Textures dataset is lower, this is mainly due to the limited overlap between the diffusion model’s distribution and the texture data. Nevertheless, GOOD tackles this by using **feature-level guidance during generation** and combining **image and feature cues during testing**. This limitation can be mitigated with more advanced generative models, and **our framework is adaptable to future advancements**.

---

> > ### Author Response · Authors · 2025-08-04
> > **Kind Follow-Up on Rebuttal**
> >
> > We are truly grateful for the time and thoughtful feedback you have provided. Our rebuttal addresses all the points raised, and we would greatly appreciate any further questions or suggestions. Your guidance is highly valued.

---

> > > ### Comment · Reviewer_Gz6x · 2025-08-06
> > >
> > > Thanks the authors for the efforts on the details response and adding test on real ODD scenario. Although I am not totally convinced by the novelty and the practical value of the proposed method, I raise my score to broadline  accept.

---

### Decision · Program_Chairs · 2025-09-17

**Decision:**

Accept (poster)

**Comment:**

This paper initially got mixed scores: two borderline rejects, one borderline accept, and one accept. The authors have submitted a rebuttal, and after considering it, all reviewers expressed satisfaction with the responses and updated their scores to borderline acceptance (score 4/5). The AC concurs with the reviewers that this paper has the advantages of strong performance, novel idea, well-written and extensive experiments. Therefore, the AC would like to recommend acceptance to this paper and encourages the authors to incorporate the clarifications and experiments provided in the rebuttal into the final version.